# MDC1 mediates Pellino recruitment to sites of DNA double-strand breaks

Mònica Torres Esteban[1,*], Matthew J Stewart[2,*], Eilis Bragginton[2], Alice Meroni[1], Annica Pellizzari[1], Alain Jeanrenaud[1], Stephen J Smerdon[2], Manuel Stucki[1]

Ubiquitylation is critically implicated in the recognition and repair of DNA double-strand breaks. The adaptor protein MDC1 mediates the recruitment of the key DNA damage responsive E3 ubiquitin ligase RNF8 to the break sites. It does so by directly interacting with RNF8 in a phosphorylation-dependent manner that involves the RNF8 FHA domain, thus initiating targeted chromatin ubiquitylation at the break sites. Here, we report that MDC1 also directly binds to two additional E3 ubiquitin ligases, Pellino 1 and 2, which were recently implicated in the DNA damage response. Through a combination of biochemical, biophysical and X-ray crystallographic approaches, we reveal the molecular details of the MDC1-Pellino complexes. Furthermore, we show that in mammalian cells, MDC1 mediates Pellino recruitment to sites of DNA double-strand breaks by a direct phosphorylation-dependent interaction between the two proteins. Taken together, our findings provide new molecular insights into the ubiquitylation pathways that govern genome stability maintenance.

## Introduction

Post-translational modifications control many aspects of DNA damage response (DDR) in mammalian cells and are crucial for genome stability maintenance. At the level of chromatin, phosphorylation of the histone H2A variant H2AX by members of the phosphoinositide-3-kinase-like protein kinase (PIKK) family is critically implicated in the recruitment of DDR factors in large nuclear condensates at sites of DNA double-strand breaks (DSBs), which are cytologically discernible as nuclear foci (Rogakou et al, 1999). Phosphorylated H2AX (termed γH2AX) spreads over megabase regions flanking the lesion site. The γH2AX-marked chromatin regions are subsequently ubiquitylated, most prominently in the H2A N-terminal tail at Lysine residues 13 and 15 (yielding H2AK13ub and H2AK15ub; Gatti et al, 2012). This ubiquitylation mark is deposited by the sequential action of the two E3 ligases, RNF8 and RNF168 (Huen et al, 2007; Kolas et al, 2007; Mailand et al, 2007; Doil et al, 2009; Stewart et al, 2009), thus flagging damaged chromatin for subsequent recruitment of the key 53BP1/Shieldin-CST and BRCA1/BARD1 DDR complexes, respectively, which regulate DSB repair pathway choice during the cell cycle (reviewed in Hustedt and Durocher [2016]). Whereas RNF168 is catalyzing ubiquitylation of the N-terminal tails of H2A-type histones (Gatti et al, 2012), RNF8 and its associated E2 ubiquitin-conjugating enzyme UBC13 promote recruitment of 53BP1/Shieldin-CST and BRCA1/BARD1 indirectly, via K63-linked ubiquitylation of H1 type linker histones at DSB sites, a chromatin mark that is subsequently read by RNF168 via its UDM1 module (Thorslund et al, 2015). RNF8 itself is recruited to sites of DSBs by the adaptor protein MDC1, which in human cells, contains four conserved Thr-Gln-X-Phe motifs (TQXF motifs, where X stands for any amino acid). In response to DSB induction, the TQXF motifs are phosphorylated on the Thr residue by the ATM kinase. This generates a binding site for the phospho-Thr–binding forkhead-associated (FHA) domain located at the RNF8 N-terminus (Huen et al, 2007; Kolas et al, 2007; Mailand et al, 2007). Because MDC1 itself is the predominant reader of the γH2AX chromatin mark, it essentially ties chromatin phosphorylation to chromatin ubiquitylation (Stucki et al, 2005; Jungmichel & Stucki, 2010).

RNF8 is not the only E3 ligase that features an FHA domain at its N-terminus. Pellino proteins are a small family of RING E3 ubiquitin ligases mainly involved in signaling events downstream of the Toll and interleukin-1 (IL-1) receptors (TIRs), which are key initiators of innate immune and inflammatory responses (Moynagh, 2014). They bind to downstream kinases recruited to TIRs (including the interleukin-1 receptor–associated kinase-1 IRAK1) and target them for ubiquitylation. In mammals, there are four Pellino proteins: Pellino 1 (PELI1), Pellino 2 (PELI2) and two splice variants of Pellino 3 (PELI3a and 3b). All four members of the Pellino E3 ligase family contain atypic FHA domains at their N-terminus with conserved key amino acids known to be important for phospho-Threonine binding by canonical FHA domains. However, in contrast to the canonical FHA domain present in RNF8, Pellino FHA domains contain two

[1]Department of Gynecology, University of Zurich and University Hospital of Zurich, Schlieren, Switzerland  [2]Institute of Cancer and Genomic Sciences, University of Birmingham, Birmingham, UK

Correspondence: s.j.smerdon@bham.ac.uk; manuel.stucki@uzh.ch
*Mònica Torres Esteban and Matthew J Stewart contributed equally to this work

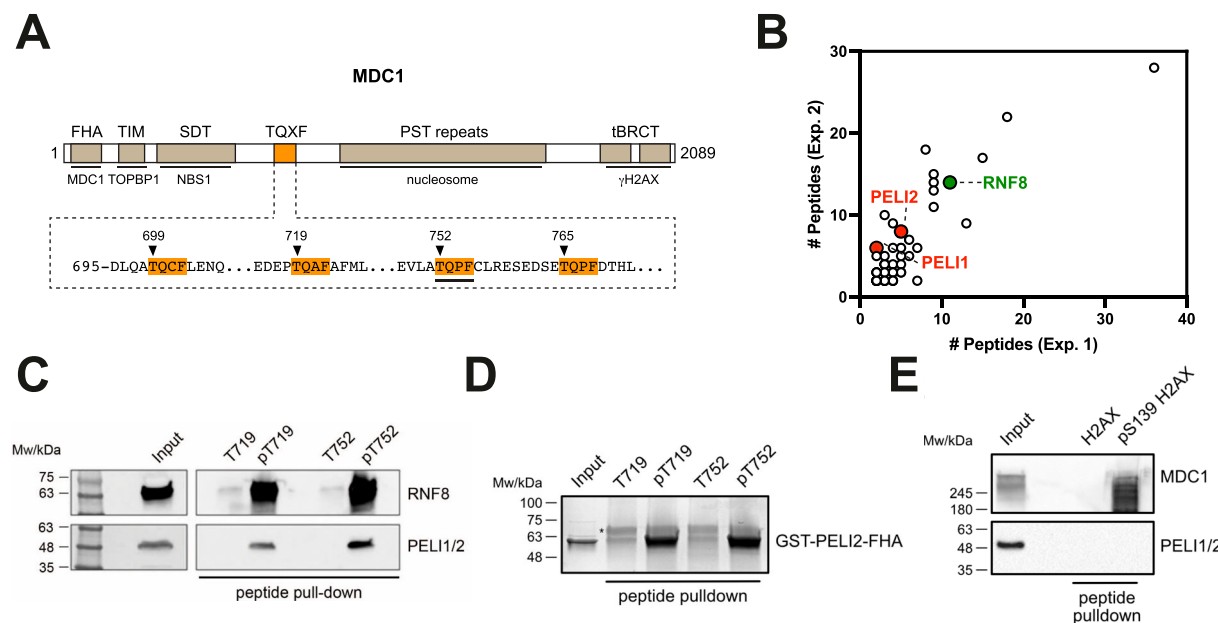

**Figure 1. MDC1 TQXF phosphopeptides interact with PELI1 and PELI2.**
**(A)** Schematic representation of MDC1 and its domain structure. The sequence of the TQXF region is shown and individual TQXF repeats are highlighted. The TQXF repeat used for peptide pull-down MS/MS is underlined. **(B)** Resulting tandem mass spectrometry (MS/MS) dataset after performing MDC1-pTQXF peptide pull-down from Hela nuclear extracts. RNF8 and the newly identified PELI1 and PELI2 proteins are highlighted. **(C)** SDS–PAGE Western blot of the MDC1-pTQXF peptide pull-down from Hela nuclear extracts showing control (non-phosphorylated) or phosphorylated biotinylated TQXF peptides (T719 and T752) and probed with RNF8 and PELI1/2 antibodies. **(D)** MDC1-pTQXF (T719 and T752) peptide pull-down from purified GST-tagged PELI2-FHA recombinant protein with control (non-phosphorylated) or phosphorylated biotinylated TQXF peptides (T719 and T752). (*) The upper band is indicated to be Streptavidin eluted from the beads. **(E)** H2AX-pS139 peptide pull-down from Hela nuclear extracts with control (non-phosphorylated) or phosphorylated biotinylated S139 peptide.

substantial sequence inserts that extend from the FHA core and coalesce to form a wing-like subdomain (Lin et al, 2008; Pennell & Smerdon, 2008).

Recently, PELI1 was implicated in the mammalian DDR. PELI1 knock-out MEFs showed signs of chromosomal instability. Moreover, in response to DSB-inducing agents, PELI1 is recruited in nuclear foci in an H2AX-dependent manner (Ha et al, 2019). Physiologically, it has been proposed that PELI1 promotes reversible ATM activation via a positive feedback mechanism that involves PELI1-dependent NBS1 ubiquitylation. In contrast to RNF8, which is recruited to sites of DNA damage by directly binding to the phosphorylated MDC1 TQXF motifs via its canonical FHA domain, it was suggested that PELI1 recruitment to sites of DSBs occurs in an MDC1-independent manner, through a direct interaction of its variant split-FHA domain with the phosphorylated H2AX C-terminus (Ha et al, 2019).

Here, by means of an unbiased proteomics screen, we found that like RNF8, PELI1, and PELI2 interact with phosphorylated TQXF peptides derived from MDC1. Biochemical characterization revealed tight and specific binding of the FHA domains of both PELI 1 and 2 paralogues to each of the MDC1 TXQF motifs but not to γH2AX. The structure of a PELI1 FHA/pTQXF complex provides for comparison with the archetypal TQXF binder, RNF8, revealing differences in molecular recognition mechanisms and providing the first insights into the role of the Pellino "wing" insert in target binding. Importantly, we also unequivocally demonstrate that in human cells, PELI1 recruitment to sites of DSBs is dependent on its direct interaction with MDC1. Taken together, our results thus suggest that

MDC1 recruits several FHA domain containing E3 ubiquitin ligases with distinct target specificities to sites of DNA damage, which is likely to have ramifications for our understanding of the ubiquitin-dependent signaling pathways that govern genome stability.

# Results and Discussion

## MDC1 TQXF phosphopeptides interact with PELI1 and PELI2

The TQXF motif is a highly conserved feature that is present in multiple copies of all vertebrate MDC1 orthologues and is even present in some non-vertebrate MDC1 orthologues such as *Drosophila* MU2. So far, the only protein reported to specifically interact with phosphorylated MDC1 TQXF sites is RNF8. This poses the question as to why most MDC1 orthologues contain several TQXF motifs and suggests the assumption that there may be additional TQXF-interacting proteins that have not yet been identified. To address this issue, we decided to search for TQXF interacting proteins by means of an unbiased proteomics approach. Human MDC1 features four TQXF repeats between amino acid residues 699 and 768 (Fig 1A). To identify proteins that bind to phosphorylated TQXF (pTQXF) motifs, we designed a synthetic biotinylated peptide corresponding to human MDC1 residues 743–757, bearing a phospho-threonine at the T752 position. This phosphopeptide and its unphosphorylated equivalent were conjugated to streptavidin-coupled beads and incubated in HeLa nuclear extracts,

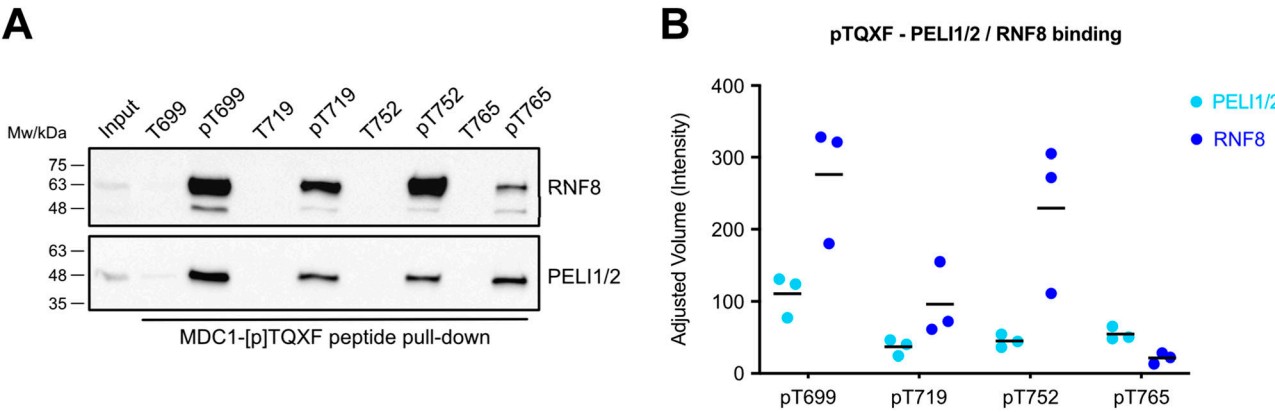

**Figure 2. PELI1/2 preferentially binds to pT699 and pT765.**
**(A)** MDC1-pTQXF peptide pull-down from Hela nuclear extracts with control (non-phosphorylated) or phosphorylated biotinylated TQXF peptides (T699, T719, T752, and T765). **(A, B)** Quantification of the experiment in (A). Black bars represent the mean PELI1/2 or RNF8 band intensity for each pTQXF peptide, from three independent experiments (n = 3).

and potential interacting partners were identified from two independent experiments by liquid chromatography-tandem mass spectrometry (LC-MS/MS; Table S1). Examination of proteins only present in the phospho-MDC1 pull-downs revealed RNF8 as a prominent interaction partner for the pTQXF motif around T752, thus serving as a positive control for this screening approach (Fig 1B). Interestingly, PELI1 and PELI2 were also present among the proteins specifically present in the phospho-MDC1 pull-downs (Fig 1B and Table S1), which, given that they both contain FHA domains that specifically recognize and bind phosphorylated peptides containing an aromatic residue at the +3 position after the pThr (Huoh & Ferguson, 2014), we decided to pursue further. Western blotting confirmed that PELI1/2 were specifically pulled down by the phosphorylated T752 peptide and also by another pTQXF phosphopeptide around T719 (Fig 1C; note that, given their extensive sequence homology, there are currently no antibodies that can distinguish between PELI1 and PELI2). To test if the interaction between PELI1/2 and the pTQXF phosphopeptides is direct and occurs via the PELI-FHA domain, we expressed and purified the PELI2-FHA domain in *E. coli* as a GST fusion protein. Both pT719 and pT752 phosphopeptides specifically pulled down the purified GST-PELI2-FHA domain whereas the native version of the peptides did not (Fig 1D). We also tested PELI1/2 binding in a γH2AX phosphopeptide pull-down. Although MDC1 was efficiently pulled down by the γH2AX phosphopeptide, PELI1/2 was not (Fig 1E). Together, these data suggest that in addition to RNF8, PELI1/2 also specifically interact with phosphorylated TQXF-containing peptides. It has been proposed that PELI1 directly interacts with the phosphorylated C-terminus of H2AX (Ha et al, 2019). This conclusion was based on co-immunoprecipitation experiments in extracts derived from cells over-expressing tagged H2AX. Because endogenous MDC1 was also present in those extracts, it is likely that the observed interaction between H2AX and PELI1 was mediated by endogenous MDC1.

## PELI1/2 preferentially binds to pT699 and pT765

To compare PELI1/2 binding preferences for the human MDC1 TQXF repeats, we designed synthetic biotinylated peptides encompassing

all four human TQXF motifs. The phosphopeptides and their unphosphorylated equivalents were conjugated to streptavidin-coupled beads, incubated in HeLa nuclear extracts, and the interacting proteins were separated by SDS–PAGE and quantified by Western blotting. pT699 peptides pulled down the highest quantities of both RNF8 and PELI1/2 (Fig 2A and B). The rest of the TQXF phosphopeptides showed significant differences in their ability to pull-down RNF8 and PELI1/2 from HeLa nuclear extracts, respectively. For example, the pT752 phosphopeptide pulled down large quantities of RNF8, but very little of PELI1/2. In contrast, the pT765 phosphopeptide pulled down more PELI1/2 than RNF8, suggesting some differences in the RNF8 and PELI1/2 binding preferences to pTQXF phosphopeptides. To quantitatively assess PELI1 and 2 binding preferences, we performed isothermal titration calorimetry (ITC) with non-biotinylated TQXF phosphopeptides and recombinant N-terminal fragments of PELI 1 and 2 (Table 1). In all cases, both PELI-FHAs bind tightly to the TQXF phosphopeptides with affinities comparable to that of other FHA domain phospho-dependent interactions. Importantly, the pT765 phosphopeptide exhibited a 5–10-fold tighter affinity than the other TQXF phosphopeptides with a $K_D$ of 230 nM, consistent with the preference for this site seen in our pull-down experiments (Figs 2B and S1A). By way of comparison, ITC titration of the N-terminal PELI1 fragment and an IRAK1-pT141(+3Y) phospho-peptide showed tight binding with an affinity (0.42 $\mu$M) comparable to that reported previously (Huoh & Ferguson, 2014) and confirming the observed +3 peptide binding preference for F/Y (Table 1). Overall, this pattern of binding preferences was recapitulated in the equivalent PELI2-FHA experiments, suggesting that, at least in this signaling context, the two paralogues are likely to be functionally redundant. Based upon these data, we asked whether PELI1 recruitment to DSBs is dependent on the phosphorylation status of MDC1 alone, or through interaction with H2AX-pS139 (γH2AX) (Ha et al, 2019). Again, we quantitatively assessed the binding of our N-terminal PELI1 and 2 fragments with a γH2AX phosphopeptide but did not detect any interaction under conditions where the γH2AX phosphopeptide was bound tightly by purified MDC1 BRCT-repeat fragment (Stucki et al, 2005; Fig S1B). This is, perhaps, not surprising given the acute pThr

**Table 1.** Isothermal titration calorimetry data.

| Peptide | Kd, uM | N | DH, kJ/mol | DG, kJ/mol | TDS, kJ/mol K |
|---|---|---|---|---|---|
| Pellino 1 FHA | | | | | |
| MDC1-pT699 (AYDLQApTQCFA) | 2.01 | 0.36 | −77.5 | −32.5 | −44.9 |
| MDC1-pT719 (AYEDEPpTQAFA) | 1.14 | 0.78 | −91.2 | −34.0 | −57.3 |
| MDC1-pT752 (AYEVLApTQPFC) | 1.14 | 0.74 | −77.2 | −33.9 | −43.3 |
| MDC1-pT765 (AYEDSEpTQPFD) | 0.23 | 0.62 | −92.1 | −38.0 | −54.1 |
| IRAK1-pT141 +3Y (AYSSASpTFLYP) | 0.42 | 0.70 | −72.6 | −36.5 | −36.1 |
| H2AX-pS139 (GGKATQApSQEY) | N.D.B. | — | — | — | — |
| Pellino 2 FHA | | | | | |
| MDC1-pT699 (AYDLQApTQCFA) | 1.12 | 0.55 | −83.4 | −34.0 | −49.4 |
| MDC1-pT719 (AYEDEPpTQAFA) | 1.22 | 0.52 | −145 | −33.8 | −111 |
| MDC1-pT752 (AYEVLApTQPFC) | 1.06 | 0.69 | −118 | −34.1 | −83.7 |
| MDC1-pT765 (AYEDSEpTQPFD) | 0.37 | 0.69 | −137 | −36.7 | −100 |
| IRAK1-pT141 (AYSSASpTFLYP) | 0.70 | 0.59 | −108 | −35.2 | −73 |
| H2AX-pS139 (GGKATQApSQEY) | N.D.B. | — | — | — | — |
| MDC1 BRCT-r | | | | | |
| H2AX-pS139 (GGKATQApSQEY) | 0.85 | 0.60 | −58.2 | −34.6 | −23.5 |

ΔG, Gibbs free energy; ΔH, change in enthalpy; ΔS, change in entropy; T, temperature.

specificity of FHA domains observed previously (Durocher et al, 2000). Our data, therefore, strongly support a mechanism, whereby PELI1 is recruited to DSB regions through direct interactions with the phosphorylated MDC1 TQXF motifs of MDC1 that is, in turn, bound to γH2AX through its BRCT-repeat domain as we described previously.

## Structure of the PELI2-FHA-MDC1-pTQXF complex

Whereas the X-ray structure of the N-terminal phospho-binding domain of PELI2 has been reported, no structures of either PELI1 or any phospho-ligand complexes have yet been determined. The X-ray structure of a recombinant N-terminal PELI1 region (residues 13–263) bound to an MDC1-pT765 phosphopeptide (PELI1$_{FHA:MDC1-pT765}$) was solved using molecular replacement with PELI2 (PDB accession code: 3EGA; Lin et al, 2008) as a search model (Table 2). Initial maps phased with just the molecular replacement model showed clear residual density for the MDC1-pT765 phosphopeptide (Fig 3A) and enabled a near-complete model of the complex to be built and refined at 2.7 Å resolution (Fig 3B). The N-terminal fragment of PELI1 consists of 16 β-strands forming four distinct β-sheets, and a single α-helix. Like the apo-PELI2 crystal structure, the PELI1 N-terminal fragment adopts an atypical FHA-fold. It features a central FHA core (residues 13–48, 97–156, and 183–257) comprising two β-sheets (β1-β2, β6-8, and β12-16) that form a β-sandwich. The FHA core is split by a bipartite "wing" appendage consisting of residues 45–96 (Wing I) and 157–182 (Wing II) consisting of the remaining two β-sheets (β3-5, β9, and β10-11) and an α-helix that separates β3-β4 and a large insertion between β3-β4. The MDC1-pT765 phosphopeptide binds in an extended conformation to a positively charged pocket within the PELI1-FHA domain, formed by loops separating β6-β7, β11-β12, and β14-β15 (Fig 3B). In total, the pT765 phosphate group forms four hydrogen-bonding interactions with side- and main-chain atoms from Arg104, Ser135, and Arg136 located in the loop separating β6-β7 of the FHA β-sandwich (Figs 3C and S2). Mutation of Arg104 to alanine has been previously shown to abrogate binding of IRAK1-pT141 phosphopeptides (Huoh & Ferguson, 2014). The phosphate contacts made by Ser135 and Arg136 are highly conserved throughout the FHA family, including in RNF8 (Ser60 and Arg61) and Chk2 (Ser140 and Lys141) DDR proteins (Li et al, 2002; Huen et al, 2007). There are an additional 12 PELI1 FHA-phosphopeptide hydrogen bonds that contribute to the ~750 Å$^2$ interaction interface. These are mediated by loops β6-β7 and β11-β12, involving Arg104, Asp116, Ser132, Arg136, Asn186, and Ser223, as well as a salt bridge formed between Arg136 and the pThr-1 Glu side chains. To our knowledge, this is the first instance of a Pellino family member interacting with a phosphopeptide that contains a Phe residue in the pT+3 peptide position, which is a well-established key determinant of binding specificity for FHA domains (Durocher et al, 2000). Previous structural analyses of PELI2 have proposed that the FHA wing appendage may contribute to binding of phospho-peptide substrates by extending the interaction surface (Lin et al, 2008). The structure now shows that the contribution of the wing appendage is indirect. The PELI1 wing domain does not make any contacts with the MDC1-pThr-765 phosphopeptide but instead acts as a structural support for a large insertion between β12-β13 (Fig 3D). This generates a shallow, positively charged platform formed predominantly by Arg 222 and Cys 247 (Fig 3D and E). This extended patch of positive electrostatic potential also appears to attract the carboxylate side chain of the pT+4 aspartate residue which is only present in the pThr-765 motif and likely contributes to the increased affinity of this site observed in the ITC experiments compared with the other three motifs that lack this feature (Fig 3E). It also provides a surface for Phe +3 binding through cation-π and Van der Waals interactions (Fig S2). This structural mechanism for +3 selectivity

**Table 2.  Crystallographic statistics.**

| Beamline | Diamond light source IO2 |
|---|---|
| Data collection/processing | |
| Wavelength, Å | 0.9795 |
| Space group | $P2_12_12_1$ |
| Cell parameters, Å | 52.9, 75.1, 165.1 |
| Cell parameters, ° | 90.00, 90, 90.00 |
| Resolution range, Å | 68.35–2.74 (2.78–2.74)[a] |
| Number of observations | 224,435 (6,165) |
| Unique reflections | 17,750 (727) |
| Completeness, % | 98.3 (80.7) |
| Mean I/I (σ) | 8.0 (0.9) |
| Multiplicity | 12.6 (8.5) |
| $R_{merge}$ | 0.236 (2.387) |
| $R_{meas}$ | 0.264 (2.449) |
| $R_{pim}$ | 0.069 (0.873) |
| $CC_{1/2}$ | 0.997 (0.407) |
| Refinement | |
| Protein atoms | 3,732 |
| Waters | 78 |
| Sulphate ions | 5 |
| $R_{cryst}$ | 0.25 (0.41) |
| $R_{free}$ | 0.27 (0.45) |
| RMSD bond-lengths, Å | 0.013 |
| RMSD bond angles, ° | 1.8 |
| Ramachandran | |
| Most favoured, % | 96.9 |
| Additional allowed, % | 3.1 |
| Disallowed, % | 0 |

[a]Highest resolution shell in parenthesis.

differs from that observed previously in RNF8, where the side chain of Phe +3 of an in vitro selected peptide resembling the MDC1 TQXF sites is rotated with respect to the PELI1 complex and, instead, bound by an insertion loop adjacent to the RNF8 FHA phospho-threonine binding pocket (Fig 3F; Huen et al, 2007).

## MDC1 and PELI1/2 interact in cells

Having established the PELI1/2-FHA-pTQXF interaction in vitro, we next investigated whether MDC1 interacted with PELI1/2 in cells and if this interaction is dependent on DNA damage. To do this, we first expressed a hemagglutinin (HA) and flag-tagged, 800 amino-acid–long N-terminal MDC1 fragment that contains all four TQXF repeats in 293T cells. In addition, a mutated derivative that had the Thr residue in all four TQXF motifs replaced by Ala (AQXF) was also expressed. Immunoprecipitation with Flag affinity beads showed that the MDC1(800) TQXF WT fragment, but not the AQXF mutant, interacted with endogenous PELI1/2. We also observed a partial

dependency of the interaction on DNA damage because, even though it was detectable in extracts prepared from control cells, the interaction was increased in extracts of cells pre-treated with 3 Gy of ionizing radiation (IR; Fig 4A).

To further test if MDC1 and PELI1/2 interact in cells in a DNA damage-dependent manner, we performed proximity ligation assays (in situ PLA). Because the commercially available PELI1/2 antibodies do not work in immunofluorescence (data not shown), we transfected U2OS cells and U2OS cells in which the *MDC1* gene was knocked-out by the CRISPR/Cas9 technology (ΔMDC1; Leimbacher et al, 2019) with an expression plasmid for full-length HA- and GFP-tagged PELI1 and tested for proximity with anti-MDC1 and anti-HA antibodies. In non-irradiated cells, only a weak PLA signal was discernible. This signal was significantly increased in irradiated cells (Fig 4B and C). Importantly, the PLA signal was strongly reduced in ΔMDC1 cells, irrespective of whether they were irradiated, indicating that it is dependent on the presence of both interaction partners (Fig 4B and C). Together, these data show that MDC1 and PELI1/2 interact in cell extracts and that in intact cells, the interaction is stimulated by irradiation. The partial DNA damage-independent interaction detected in cell extracts by co-immunoprecipitation is surprising because in the absence of DNA damage, the TQXF motifs are not expected to be phosphorylated. However, previous work showed that RNF8 and MDC1 also co-immunoprecipitated in unstressed cells (Huen et al, 2007; Kolas et al, 2007; Mailand et al, 2007) and that MDC1 was phosphorylated to some extent even in non-irradiated cells (Mailand et al, 2007). A possible explanation for this may be a weak intrinsic ATM/ATR activity and/or induction of the DDR by the transfection of linearized plasmid DNA. Consistent with this, there is also a weak PLA signal detectable in unirradiated cells.

## MDC1 TQXF–dependent recruitment of PELI1 to sites of DSBs

To test if MDC1 is required to recruit PELI1/2 to sites of DSBs in mammalian cells we initially attempted to use commercial anti-bodies raised against PELI proteins in immunofluorescence (IF). Whereas these antibodies work relatively well in Western blotting (see above), they did not yield a specific signal in IF. Thus, we had to resort to recombinant expression of GFP-tagged full-length human PELI1. First, we stably transfected U2OS cells and U2OS ΔMDC1 cells with a PELI1 expression vector. These cells were pre-incubated with the halogenated nucleotide analoga BrdU for 24 h and then subjected to microlaser irradiation on a microdissection microscope coupled to a pulsed UVA laser. The cells were subsequently fixed and stained with anti-γH2AX and anti-MDC1 antibodies. In ΔMDC1 cells, no GFP-PELI1 signal was observed in γH2AX-positive micro-laser tracks, whereas in control U2OS cells, the GFP-PELI1 signal was enriched in the microlaser tracks, where it co-localized with the MDC1 signal (Fig 5A). To test if the TQXF cluster is required for PELI1 DSB recruitment, we co-transfected ΔMDC1 cells with GFP-tagged full-length MDC1 and red fluorescent protein (RFP) tagged PELI1. In the case of MDC1, we either used a WT expression construct or an AQXF mutant, having the Thr residue in all four TQXF motifs mutated to Ala. These cells were subjected to the same microlaser treatment as described above. Both WT MDC1 and the AQXF mutant were efficiently recruited to the γH2AX marked laser tracks. However, no

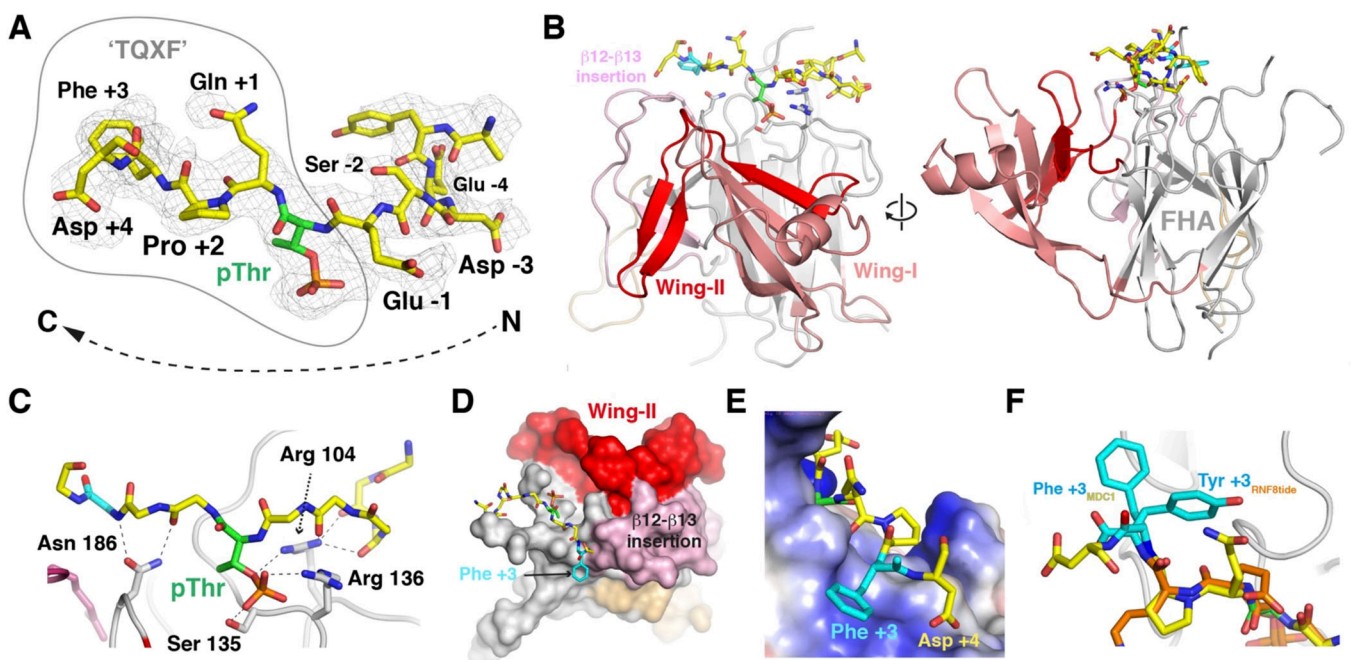

**Figure 3. Structure of the PELI2-FHA-MDC1-pTQXF complex.**
**(A)** Difference electron density map contoured at 1σ clearly showing the presence of a bound phosphopeptide. The final, refined peptide structure is superposed. **(B)** Orthogonal views of the PELI1 split-FHA domain in complex with the MDC1-pT765 motif. The bipartite wing structure is shown along with a second large insertion between β12-β13. **(C)** Hydrogen-bonding interactions between the MDC1 p765 phosphopeptide and the four most highly conserved FHA domain residues that mediate similar interactions in other systems. **(D)** Surface representation of the PELI1 FHA domain showing how Wing II buttresses the β12-β13 insertion loop to create a shallow pocket for the Phe +3 residue. **(E)** Electrostatic potential surface at the +3 binding pocket. Positive charge is largely supplied by the underlying Arg 222 residue which engages the benzyl side chain of Phe +3 and the carboxylate of the Asp +4 position. **(F)** Overlap of the MDC1 p765 peptide from the PELI1 complex with the in vitro selected RNF8 peptide reveals differences in the specificity-determining interactions with the +3 side chains in each case.

RFP-PELI1 signal enrichment was observed in cell lines expressing the MDC1-AQXF mutant (Fig 5B). These data clearly indicate that PELI1 recruitment to sites of microlaser-induced DSBs is dependent on MDC1 and, more specifically, on the intact TQXF motifs in MDC1, containing phosphoacceptor Thr residues. To test if MDC1 TQXF motifs were also required to recruit PELI1 in IR-induced foci, we subjected the transfected GFP-MDC1/RFP-PELI1 expressing ΔMDC1 cell lines to 3 Gy of IR and stained them with anti-γH2AX antibodies. Whereas RFP-PELI1 accumulated in clearly discernible foci that co-localized with γH2AX and GFP-MDC1 foci in WT GFP-MDC1-WT expressing cells, no such accumulation was observed in the GFP-MDC1-AQXF expressing cells (Fig 5C). Quantitative assessment of RFP-PELI1 foci formation by computer-aided segmentation and quantification of subcellular shapes (SQUASSH; Rizk et al, 2014) revealed a significant degree of colocalization between GFP-MDC1 and RFP-PELI1 in the GFP-MDC1-WT expressing cell line. However, this colocalization is almost completely abrogated in GFP-MDC1-AQXF expressing cells, indicating that the intact TQXF motifs are required to recruit PELI1 to foci at sites of DSBs (Fig 5D). Interestingly, we also observed that GFP-RNF8 could co-localize with RFP-PELI1 and with MDC1 in IR-induced foci, indicating that PELI1 and RNF8 can simultaneously be recruited to sites of DSBs and that Pellinos and RNF8 do not compete for binding to the phosphorylated MDC1 TQXF repeats, at least not at the expression levels tested (Fig S3).

It was reported that PELI1 is recruited to sites of DSBs in a γH2AX-dependent but MDC1-independent manner (Ha et al, 2019). This is at odds with our data. The conclusion of Ha et al was based on an experiment in which endogenous MDC1 was knocked down by siRNA transfection and GFP-PELI1 recruitment to sites of DSBs was monitored after microlaser irradiation of the siRNA transfected cells. A closer inspection of the efficiency of the siRNA depletion showed that the MDC1 was only partially knocked down, with a significant proportion of the protein still visible on the Western blot (Ha et al, 2019). Inefficient MDC1 down-regulation may thus be a conceivable explanation for the data discrepancy. In summary, our data suggest that PELI1, like RNF8, is recruited to sites of DNA damage by direct interaction with the phosphorylated TQXF motifs in MDC1. We and others identified six conserved acidic sequence motifs upstream of the TQXF repeats in MDC1 that directly interact with the N-terminal FHA/BRCT region in NBS1 and are required for NBS1 foci formation at sites of DSBs (Chapman & Jackson, 2008; Melander et al, 2008; Spycher et al, 2008). It was proposed that the physiological targets of the PELI1 E3 ligase at sites of DSBs are lysine residues K686 and K690 of NBS1, a subunit of the MRE11/RAD50/NBS1 (MRN) complex (Ha et al, 2019). It is thus conceivable that these NBS1 lysine acceptor sites may be juxtaposed with the PELI1 RING domain through combined phospho-dependent interactions of NBS1 and PELI1/2 with distinct but highly conserved clusters of phosphosites within the MDC1 scaffold.

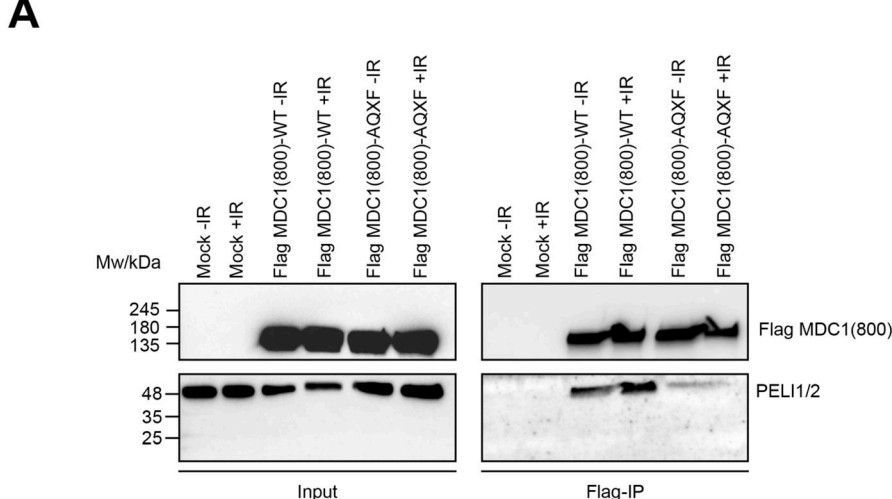

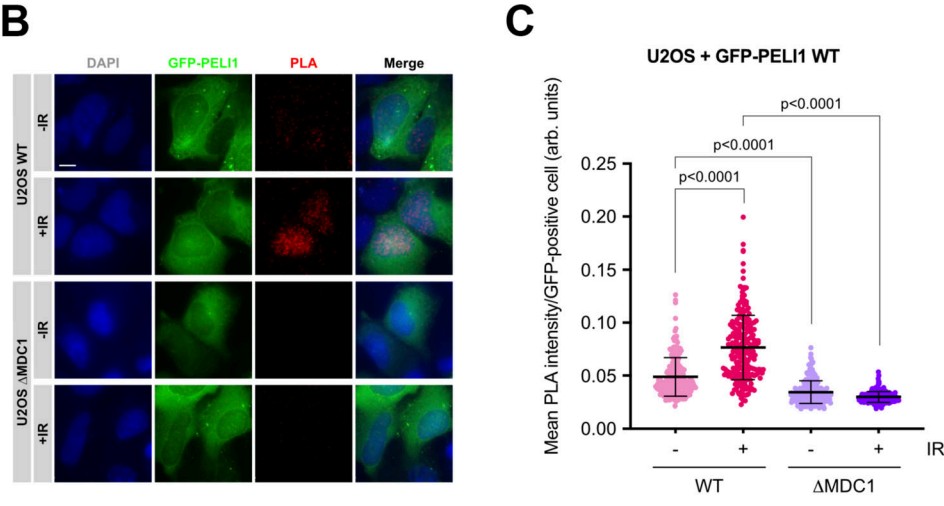

**Figure 4. MDC1 and PELI1/2 interact in cells.**
**(A)** SDS–PAGE Western blot of the Flag-tag co-immunoprecipitation experiment of endogenous proteins using HA/Flag-MDC1(800aa) fragment (WT and AQXF mutant), showing input and immunoprecipitated samples, together with the different transfection and irradiation conditions, as well as the blotting for HA-tag (Flag-MDC1(800aa)) and PELI1/2 with their respective sizes. **(B)** Widefield micrographs (maximum intensity projections) of in situ PLA, using antibodies against MDC1 and HA (Strep-HA-GFP-PELI1) to detect interaction between MDC1 and GFP-PELI1. U2OS WT and ΔMDC1 cells were transiently transfected with GFP-PELI1 and processed 3 h after irradiation with 3 Gy (+IR) or without treatment (−IR). **(A, C)** Quantification of MDC1-GFP-PELI1 proximity by in situ PLA in cells from (A). Each data point represents one cell (n = 229; one representative of two independent experiments is shown). Bars and error bars represent mean and SD. Statistical significance was assessed by the two-tailed unpaired $t$ test with Welch's correction ($\alpha = 0.05$). All scale bars = 10 $\mu$m.

# Materials and Methods

## Cell lines, cell culture conditions, and treatments

All cell lines were grown in a sterile cell culture environment and were routinely tested for mycoplasma contamination. U2OS (U2OS, human bone osteosarcoma cell line; ATCC) cells were cultured in DMEM. All cell culture medium was supplemented with 10% FBS, 2 mM L-glutamine and penicillin-streptomycin antibiotics. Cell lines were grown under standard cell culture conditions in $CO_2$ incubators (37°C; 5% $CO_2$). Genetically modified U2OS cell lines U2OS ΔMDC1, U2OS ΔMDC1 + GFP-MDC1 WT and U2OS ΔMDC1 + GFP-MDC1-AQXF were described (Leimbacher et al, 2019). Stably transfected U2OS cell lines were cultured in the presence of 400 $\mu$g/ml Geneticin (G418). Irradiation of cells was performed in an YXLON, Y.SMART583 custom-made X-ray device and an IR dose of 3 Gy followed by 3 h of incubation at 37°C was applied. Laser micro-irradiation of cells was carried out using a MMI CELLCUT system containing an ultraviolet (UV) A laser of 355 nm where energy output

was set to 50%, and each cell was exposed to laser beam for <300 ms, following incubation for 30 min at 37°C (described in Eid et al [2010]).

## Cloning

To generate the GFP-PELI1 expression construct, the full-length PELI1 cDNA sequence was synthesized by BioCat GmbH and pcDNA4/TO-Strep-HA-GFP-Treacle plasmid, which was described (Larsen et al, 2014), was digested by KpnI and ApaI restriction enzymes to delete Treacle's coding sequence. PCR with Phusion High-Fidelity DNA Polymerase was performed to amplify PELI1 cDNA sequence with primers partially overlapping the cohesive ends of the above-mentioned digested pcDNA4/TO vector. Sequences of these PCR primers were, forward: 5′-ATGAGCTGTA-CAAGGGTACCATGTTTTCTCCTGATCAAG-3′, and reverse: 5′-CAGCGGGTT TAAACGTCTAGATTAGTCTAGAGGTCCTTGA-3′. The amplified PELI1 cDNA sequence was cloned into the KpnI-ApaI-digested pcDNA4/TO-Strep-HA-GFP vector by In-Fusion HD Cloning (Takara) to

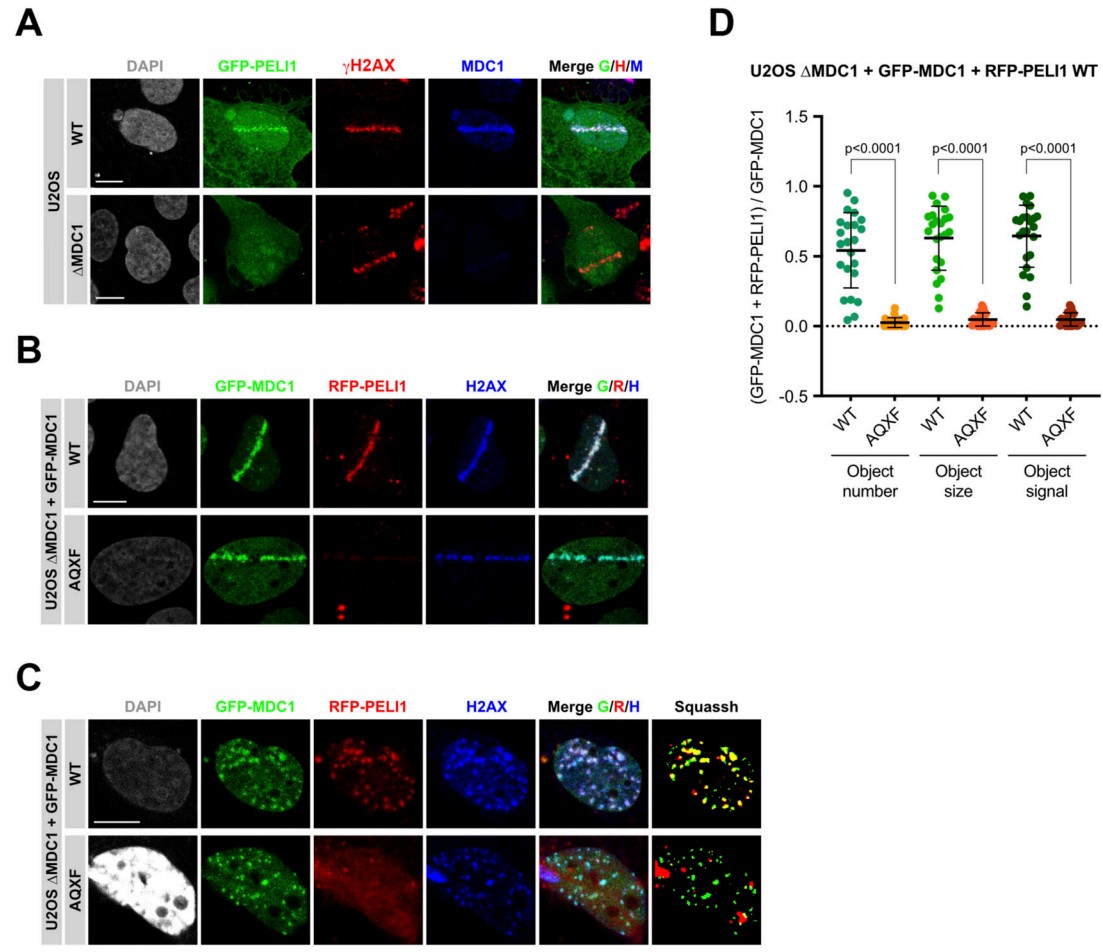

**Figure 5.    MDC1 TQXF-dependent recruitment of PELI1 to sites of double-strand breaks.**
**(A)** Confocal micrographs of GFP-PELI1 transfected U2OS WT and ΔMDC1 cells, 30 min after laser microirradiation at 50% of energy output and for <300 ms each cell, stained for γH2AX and MDC1. Displayed are maximum intensity projections of confocal z-stacks. **(B)** Confocal micrographs (maximum intensity projections) of U2OS ΔMDC1 cells stably expressing either GFP-MDC1 WT or GFP-MDC1-AQXF and transiently transfected with RFP-PELI1, 30 min after laser microirradiation at 50% of energy output and for <300 ms each cell, stained for γH2AX. **(C)** Confocal micrographs (maximum intensity projections) of U2OS ΔMDC1 cells stably expressing GFP-MDC1 WT or GFP-MDC1-AQXF and transiently transfected with RFP-PELI1, 3 h after ionizing radiation with 3 Gy and stained for γH2AX. Last panels on the right are micrographs deconvoluted and segmented by SQUASSH. **(C, D)** Quantitative analysis of GFP-MDC1 and RFP-PELI1 colocalization in cells from (C) by SQUASSH. Left: object number colocalization (fraction of objects in each channel that overlap ≥ 50%). Middle: object size colocalization (area of object overlap divided by total object area). Right: object signal colocalization (sum of all pixel intensities in one channel in all regions where objects overlap). Each data point represents one cell (n = 23; pooled from two independent experiments). Bars and error bars represent mean and SD. Statistical significance was assessed by the two-tailed unpaired Mann-Whitney test (α = 0.05). All scale bars represent 10 µm.

produce plasmid pcDNA4/TO-Strep-HA-GFP-PELI1. The construct was transformed into Stellar Competent *E. coli* cells (Takara) for further colony sequence verification by Sanger sequencing and plasmid DNA extraction. To generate the RFP-PELI1 expression construct, the RFP coding sequence was amplified by PCR from vector pTurboRFP-B (Evrogen), with Phusion High-Fidelity DNA Polymerase (NEB) using the following primers, forward: 5′-AGCA-CAGTGGCGGCCATGAGCGAGCTGATCAAGGA-3′, and reverse: 5′-GAGAA AACATGGTACTCTGTGCCCCAGTTTGCTAGG-3′. The previously pro-duced pcDNA4/TO-Strep-HA-GFP-PELI1 plasmid was digested with NotI-HF (NEB) and KpnI-HF (NEB) restriction enzymes to delete the GFP coding sequence. The amplified RFP sequence was cloned into the NotI-KpnI–digested pcDNA4/TO-Strep-HA-PELI1 vector by In-Fusion Snap Assembly (Takara) to produce plasmid pcDNA4/TO-Strep-HA-RFP-PELI1. The construct was transformed into Stellar

Competent cells for sequence verification by Sanger sequencing and further plasmid DNA extraction. To generate the GST-PELI2-FHA expression construct, the full-length PELI2 cDNA sequence was synthesized by BioCat GmbH. The FHA domain of the PELI2 protein (comprising aminoacids 1–275) was amplified by PCR from the full-length cDNA sequence with Phusion High-Fidelity DNA Polymerase using the primers forward: 5′-CGCGGATCCATGTTTTCCCCTGGCCAG-GAGGAAC-3′, and reverse: 5′-CCGGAATTCCTGCCGGAGGGCTTC-TATGTGCTTC-3′, which contained BamHI and EcoRI restriction sites, respectively, as overhangs. The PCR product was digested with BamHI-HF (NEB) and EcoRI (NEB) restriction enzymes as well as the bacterial expression vector pGEX-4T-3 (containing the GST-tag), followed by ligation with T4 DNA Ligase (NEB). This resulted in the GST-PELI2-FHA construct, which was transformed into Stellar Competent cells for Sanger sequencing and plasmid DNA

extraction. To generate the HA-Flag-MDC1(800)-AQXF expression construct, PCR with Q5 High-Fidelity 2X Master Mix (NEB) was performed using 15-nucleotide overlapping primers to amplify the AQXF mutant sequence (forward: 5′-GAAGACAACTATGGTGATTCTGAA GATCTGGACCTACAAG-3′, and reverse: 5′-AGGTGGAGACAGGCAAGGTC CATAGGCCTCAAGGTG-3′) and the plasmid backbone sequence (forward: 5′-TGCCTGTCTCCACCTAGGG-3′, and reverse: 5′-ACCA-TAGTTGTCTTCAGAGTCC-3′), deleting the WT TQXF cluster in the process. These original expression plasmids were pcDNA3.1-GFP-MDC1-AQXF which was a gift from Stephen Jackson's laboratory and was described elsewhere (Kolas et al, 2007), and pcDNA3.1-HA-Flag-MDC1(800)-WT which was described in Jungmichel and Stucki (2010). The two resulting DNA fragments were then fused by In-Fusion Snap Assembly to produce plasmid pcDNA3.1-HA-Flag-MDC1(800)-AQXF. The construct was transformed into Stellar Competent cells for sequence verification by Sanger sequencing and further plasmid DNA extraction.

### Cell transient transfection with DNA

Plasmid DNA was transiently transfected using jetOPTIMUS DNA Transfection Reagent (Polyplus) in six-well plates following the standard protocol provided by the manufacturer. Cells were then incubated for 24 h before proceeding with further analysis.

### Expression and purification of recombinant proteins for biochemical analysis

For recombinant human PELI2-FHA expression in bacteria and further purification, the GST-tagged PELI2-FHA construct was transformed into E. coli BL21(DE3)pLysS Competent Cells (Promega). 100 $\mu$l of the transformation mixture were used to inoculate a 500 ml flask containing 100 ml of Luria–Bertani broth (LB) supplemented with carbenicillin (100 $\mu$g/ml) and chloramphenicol (34 $\mu$g/ml), and inoculated culture was incubated overnight at 27°C without shaking. On the next day, the overnight culture was grown at 37°C in an orbital shaking incubator until the optical density at 600 nm reached 0.6–0.8, moment when culture was removed from the incubator and rapidly cooled on ice to 25°C. Recombinant protein expression was induced by the addition of 0.4 mM isopropyl $\beta$-D-1-thiogalactopyranoside (IPTG) and incubation of the culture for a further 4 h at 25°C shaking at 220 rpm was performed. Cells were then harvested by centrifugation, and the resulting pellet was stored at –80°C until required. Bacterial pellet was lysed through repeated freezing (down to –80°C) and thawing and by resuspension in 5 ml of ice-cold lysis/wash buffer (25 mM HEPES pH 7.5, 1 M NaCl, 5% glycerol) supplemented with complete protease inhibitor cocktail (Roche), followed by vigorous sonication on ice delivered by a Jencons Ultrasonic Processor (15 s bursts at 70% amplitude). Triton X-100 was added to the lysate to a final concentration of 1% before incubation for 45 min rolling at 4°C and clearance by centrifugation for 30 min at 15,500 rcf and 4°C. After this step, the cleared extract was loaded onto a 5 ml Bio-Scale Mini Profinity GST Cartridge (Bio-Rad) for GST affinity purification on an NGC Quest 10 chromatography system (Bio-Rad). For this run, lysis/wash buffer and elution buffer pH 7.5 (25 mM HEPES pH 7.5, 1 M NaCl, 5% glycerol, 30 mM Glutathion) were used. Expression and

purification levels of the purified protein fraction were tested on SDS–PAGE followed by gel staining with QuickBlue Protein Stain (LubioScience) for protein visualization, and fractions with reasonable protein content and purity were pooled. Dialysis of pooled fractions was performed overnight at 4°C with slow stirring against Buffer A (50 mM Tris–HCl pH 7.5, 50 mM NaCl, 10% glycerol, 1 mM DTBA). Next, pooled and dialysed fractions were loaded onto a 1 ml ENrich Q 5 × 50 Column (Bio-Rad) for ion-exchange purification on the NGC system (Bio-Rad). For this second run, Buffer A and Buffer B (50 mM Tris–HCl pH 7.5, 1 M NaCl, 10% glycerol, 1 mM DTBA) were used. Finally, the purified protein fraction was collected. Protein concentration was measured after every purification step.

### Expression and purification of recombinant proteins for biophysical and X-ray structure analysis

The regions corresponding to the PELI1 and 2 FHA domains (residues 13–273 and 15–275, respectively) were cloned into the pGEX-6P1 vector using BamHI/XhoI restriction sites and expressed as a 3C protease-cleavable N-terminal GST fusion protein in BL21 (DE3) cells (NEB). A 20 ml LB overnight culture was inoculated into 1L of LB medium supplemented with 100 $\mu$g/ml ampicillin to an OD600 of 0.05 and incubated at 37°C, 250 rpm until an OD600 of 0.6 was reached, whereupon expression was induced with 1 mM isopropyl ß-D-1-thiogalactopyranoside (IPTG) and overnight incubation at 18°C. Overnight cultures were harvested for 20 min at 4,000$g$, 4°C and resuspended in 20 ml lysis buffer (50 mM HEPES pH 7.5, 150 mM NaCl, 0.2 mg/ml lysozyme, 20 $\mu$g/ml DNase, and 1x complete, EDTA-free protease inhibitor cocktail tablet) before sonication-mediated cell lysis. Insoluble material was removed by centrifugation (21,000$g$, 20 min) and the GST-FHA fusion proteins were purified from the clarified lysate containers using 1 ml bed volume of pre-equilibrated Glutathione Sepharose 4B beads. After washing with 20 column volumes of Pellino FHA buffer (50 mM HEPES pH 7.5, 150 mM NaCl), the protein was cleaved overnight at 4°C from GST with 1% (vol/vol) GST-3C protease and the flow-through was further purified by gel filtration using a HiLoad 16/60 Superdex 200 pg (Cytiva) column. Purified Pellino FHA domains were concentrated to 50 mg/ml with a 10 kD molecular weight cut-off concentrator (Sartorius).

### ITC

For ITC analysis protein and peptides were buffer exchanged into 25 mM HEPES pH 7.0, 200 mM NaCl, 1 mM TCEP. Titrations were carried out using a Microcal PEAQ-ITC (Malvern) using 13 injections of peptide at a typical syringe concentration of 160 $\mu$M (1 × 0.4 $\mu$l, 12 × 3.0 $\mu$l) and FHA concentration of 16 $\mu$M in the cell. Data were analyzed with a single-site binding model using Microcal PEAQ-ITC software. Peptide sequences are: MDC1 pT699–AYDLQApTQCFA, MDC1-pT719–ATEDEPpTQAFA, MDC1-pT752–AYEVLApTQPFC, MDC1-pT765–AYEDSEpTQPFD, IRAK-pT141(+3Y)–AYSSASpTFLYP, γH2AX–GGKATQApSQEY. All peptides were synthesises with N-terminal acetylation and a C-terminal amide. MDC1 pT699 and pT719 were made with alanine substitutions at the pT+4 position (Leu and Met, respectively) to aid solubility and monodispersity.

## Crystallographic analysis

For crystallisation, the MDC1-pT765 phosphopeptide with the Pellino 1FHA fragment in a 1,170 μM:754 μM stoichiometric excess in Pellino FHA buffer. Crystallisation using the sitting drop vapour diffusion method was conducted at 18°C and was achieved by mixing the 200 nl of the Pellino FHA:MDC1 phosphopeptide complex in a 1:1 ratio with 200 nl of mother liquor. Initial trials were performed using the Proplex crystal screen (Molecular Dimensions) and crystals grew after 1 wk in 1.5 M ammonium sulphate, 0.1 M Tris pH 8. Additional trials were performed using the Ångstrom Additive Screen (Molecular Dimension) in a 1:9 ratio with 1.67 M ammonium sulphate, 0.11 M Tris pH 8.0, whereby single crystals grew from drops consisting of 1.5 M ammonium sulphate, 0.1 M Tris pH 8, 20% (vol/vol) PEG400. Crystals were cryoprotected by soaking in mother liquor supplemented with 25% (vol/vol) PEG400. Diffraction data were collected at the Diamond Light Source beamline I04 at a wavelength of 0.9795 Å and processed using autoPROC (Vonrhein et al, 2011), and the structure was phased by molecular replacement using PHASER from the CCP4i suite with a search model comprised of coordinates from the human Pellino 2FHA structure (PDB accession code 3EGA; Lin et al, 2008). The model was manually rebuilt and adjusted in Coot and refined using PHENIX (Table 2). Coordinates for the Pellino-phosphopeptide complex have been deposited with the Protein Data Bank: Accession code 9IF9.

## Peptide pull-downs

Streptavidin-coupled Dynabeads MyOne Streptavidin T1 (Invitrogen) were washed four times with PBS pH 7.4. To conjugate the Dynabeads with biotinylated peptides (21st Century Biochemicals, Inc.), 30 μl of beads were incubated with 0.2 μmol of each peptide with end-over-end rotation for 90 min at RT. Conjugated Dynabeads were then washed once with PBS and 3 × 2 min with 0.1% BSA in PBS. HeLa nuclear extracts (Ipracell) were diluted 1:1 in 2X Dilution buffer (100 mM NaCl, 10 mM NaF, 0.2 mM EDTA, 0.4% Igepal CA-630, 20 mM HEPES-KOH pH 7.4) supplemented with complete protease inhibitor cocktail (Roche), and cleared by centrifugation at 16,000g for 15 min at 4°C. Purified GST-PELI2-FHA recombinant protein was also diluted in 2X Dilution buffer containing protease inhibitors. MDC1-[p]TQXF peptide-conjugated Dynabeads were incubated either with clarified HeLa nuclear extracts for 4 h or with purified GST-PELI2-FHA for 5 h. H2AX-[p]S139 peptide-conjugated Dynabeads were incubated with clarified HeLa nuclear extracts for 4 h. In all the cases, incubation was performed with end-over-end rotation at 4°C. Beads were washed three times with Peptide pull-down buffer (50 mM NaCl, 50 mM KCl, 5 mM NaF, 0.1 mM EDTA, 0.2% Igepal CA-630, 10% glycerol, 20 mM HEPES-KOH pH 7.4) supplemented with complete protease inhibitor cocktail (Roche). Then, they were either resuspended in Peptide pull-down buffer and brought to the Functional Genomics Center of the University of Zurich (FGCZ) for further mass spectrometry, or elution was carried out in 2X SDS sample buffer (100 mM DTT, PLD001; Geneaid) for SDS–PAGE. Peptide sequences were as follows: Biotin-SGS-DSEDLDLQA[p]TQCFLE-NH₂ (MDC1-[p]T699), Biotin-SGS-AVQSMEDEP[p]TQAFML-NH₂ (MDC1-[p]T719), Biotin-SGS-LDEPWEVLA[p]TQPFCL-NH₂ (MDC1-

[p]T752), Biotin-SGS-CLRESEDSE[p]TQPFDT-NH₂ (MDC1-[p]T765), Biotin-SGS-TVGPKAPSGGKKATQA[p]SQEY-OH (H2AX-[p]S139).

## Immunoprecipitations

293FT cells were grown in six-well plates overnight and then transfected with pcDNA3.1-HA-Flag-MDC1(800) WT or AQXF mutant plasmids. After 24 h, cells were treated with 3 Gy of IR as described above. To prepare cell extracts for Flag immunoprecipitation, cells were washed in cold PBS buffer pH 7.45, harvested with a cell scraper and centrifuged at 500g for 3 min. PBS was then discarded and cells were resuspended with an appropriate volume of IP lysis buffer (50 mM Tris–HCl pH 7.4, 150 mM NaCl, 1 mM EDTA, 1% Triton X-100) supplemented with complete EDTA-free protease inhibitor cocktail (Roche) and 25 U/ml Benzonase nuclease (Sigma-Aldrich). Cells were incubated on ice for 30 min and vortexed every 10 min to dissolve cell clumps. After incubation, lysates were cleared by centrifugation at 13,200 RPM for 15 min at 4°C and transferred into fresh tubes for protein quantification. Anti-FLAG M2 affinity gel (A2220; Sigma-Aldrich) was added to fresh tubes, centrifuged at 5,000g for 30 s and washed 3 × 2 min in cold TBS (50 mM Tris–HCl pH7.4, 150 mM NaCl) with end-over-end rotation at 4°C. Then, 20 μl of packed gel were added to 2 mg of the soluble cell extract and samples were incubated for 2 h with end-over-end rotation at 4°C. After incubation, the unbound fraction was discarded, and immunoglobulin-antigen complexes were washed 3 × 15 min in cold TBS with end-over-end rotation at 4°C. After the last wash, protein complexes were eluted using 50 μl of 2X SDS sample buffer (100 mM DTT; Geneaid), denatured at 95°C for 5 min, and vortexed before loading on the gel.

## Mass spectrometry

MDC1-pTQXF and H2AX-pS139 peptide pull-downs from HeLa nuclear extracts were performed as described above, followed by mass spectrometry analysis. After the last wash, precipitated material was resuspended in peptide pull-down buffer and subjected to on-bead trypsin digestion according to the following protocol. Dynabeads were washed twice with 50 μl of digestion buffer (10 mM Tris, 2 mM CaCl₂, pH 8.2). Then, the buffer was discarded and beads were resuspended in 45 μl of digestion buffer supplemented with 5 μl of trypsin (100 ng/μl in 10 mM HCl). pH was adjusted to 8.2 by adding 1 M Tris pH 8.2. Digestion was carried out at 37°C for 5 h shaking at 800 rpm. Supernatants were collected, and peptides were extracted from beads using 150 μl of 0.1% trifluoroacetic acid (TFA) and 50% acetonitrile (ACN). Digested samples were dried and reconstituted in 20 μl ddH2O + 0.1% formic acid before performing liquid chromatography-mass spectrometry analysis (LC-MS/MS). For the analysis, 1 μl was injected on a nanoACQUITY UPLC system coupled to a Q-Exactive mass spectrometer (Thermo Fisher Scientific). MS data were processed for identification using the Mascot search engine (Matrixscience), and the spectra were searched against the Swissprot protein database. Protein and peptide identification results were visualized in Scaffold Proteome Software (Proteome Software, Inc., https://www.proteomesoftware.com/products).

## SDS–PAGE and Western blotting

SDS–PAGE of the peptide pull-down samples and the immuno-precipitated samples was performed using 4–20% Mini-PROTEAN TGX Stain-free Protein Gels (Bio-Rad). As a reference to molecular weights, Prestained Protein Ladder 245 kD (Geneaid) was also loaded on the gel. SDS–PAGE of the MDC1-[p]TQXF peptide pull-down from purified GST-PELI2-FHA recombinant protein was followed by gel staining with QuickBlue Protein Stain (LubioScience) for protein visualization. SDS–PAGE of the MDC1-[p]TQXF and H2AX-[p]S139 peptide pull-downs from HeLa nuclear extracts and the Flag immunoprecipitation were followed by Western blotting, which was performed using the Trans-Blot Turbo Transfer System (Bio-Rad) on 0.2 $\mu$m nitrocellulose membranes (Bio-Rad). Membranes were blocked for at least 2 h with 5% milk in TBS-Tween. Primary antibodies were diluted in 2.5% milk in TBS-Tween and incubation was performed at 4°C overnight. Blots were then washed 3 × 5 min with TBS-Tween. Secondary antibodies were diluted in 2.5% milk in TBS-Tween and incubation was performed for 1 h at RT. Blots were washed again 3 × 5 min with TBS-Tween and then developed with Amersham ECL Prime Western blotting Detection Reagent (Cytiva), and image acquisition was performed on a ChemiDoc MP Imaging System (Bio-Rad). The following primary antibodies were used at the indicated dilutions: anti-RNF8 (1:5,000; Rabbit), anti-PELI1/2 (Rabbitpolyclonal, 1:1,000, ab13812; Abcam), anti-MDC1 (Rabbit polyclonal, 1:5,000, ab11171; Abcam), anti-flag M2 (Mouse monoclonal, 1/1,000, F1804; Sigma-Aldrich). The following secondary antibodies were used: ECL HRP-conjugated donkey anti-rabbit (1:10,000, NA934; Cytiva), ECL HRP-conjugated sheep anti-mouse (1:10,000, NA931; Cytiva).

## Immunofluorescence

U2OS WT and ΔMDC1 cells were grown on glass coverslips (Thermo Fisher Scientific Menzel) overnight and then transfected with pcDNA4/TO-Strep-HA-GFP-PELI1 plasmid. After 24 h, coverslips were transferred into two-well chamber imaging slides (Ibidi) and treated with 10 $\mu$M 5-bromo-2'-deoxyuridine (BrdU) for 24 h before being subjected to laser microirradiation as described above. U2OS ΔMDC1 cells stably expressing either GFP-MDC1 WT or GFP-MDC1-AQXF were grown on glass coverslips overnight and then transfected with pcDNA4/TO-Strep-HA-RFP-PELI1 plasmid. After 24 h, cells were either treated with BrdU and laser microirradiation on Ibidi slides or with ionizing radiation as described above. Microlaser-treated cells were then washed twice with PBS and irradiated cells were washed once with PBS, before fixation with 4% buffered formaldehyde for 12–15 min at RT. Formaldehyde was discarded and cells were washed three times with PBS at RT, subsequently permeabilized for 5 min with 0.3% Triton X-100 in PBS, washed again three times with PBS and incubated with blocking buffer (10% FBS in PBS) for at least 1 h at RT. Primary antibodies were diluted in 10% FBS in PBS and incubation was performed at 4°C overnight. Coverslips were then washed 3 × 10 min with PBS. Secondary antibodies were diluted in 5% FBS in PBS and incubation was performed for 1 h at RT in the dark. After washing twice with PBS, coverslips were mounted on glass microscopy slides (Thermo Fisher Scientific) with VECTASHIELD PLUS Antifade Mounting

Medium containing 0.5 $\mu$g/ml DAPI (Vector Laboratories). For the GFP-PELI1-transfected U2OS WT and ΔMDC1 cells, the following primary antibodies were used at the indicated dilutions: anti-γH2AX (Mouse monoclonal, 1:500, 05-636-I, RRID: AB_2755003; Sigma-Aldrich), anti-MDC1 (Rabbit polyclonal, 1:200, ab11171, RRID: AB_297810; Abcam). Also, these secondary antibodies were used: Alexa Fluor 568 Goat Anti-Mouse IgG H&L (1:1,000, A11031, RRID: AB_144696; Thermo Fisher Scientific), Alexa Fluor 700 Goat Anti-Rabbit IgG H&L (1:500, A21038; RRID: AB_2535709; Thermo Fisher Scientific). For the RFP-PELI1-transfected U2OS GFP-MDC1 WT and U2OS GFP-MDC1-AQXF cells, the following primary antibody was used at the indicated dilution: anti-γH2AX (Rabbit monoclonal, 1:400, 9,718; Cell Signaling Technology), and this secondary antibody: Alexa Fluor 700 Goat Anti-Rabbit IgG H&L.

## In situ proximiy-ligation assay (PLA)

U2OS WT and ΔMDC1 cells were grown on glass coverslips in six-well plates at a density of 2 × 10$^5$ cells per well (and two coverslips per well) for transfection with pcDNA4/TO-Strep-HA-GFP-PELI1 plasmid on the next day. After 24 h, coverslips were transferred into 24-well plates and subjected to ionizing radiation as described above. Cells were then washed once with PBS for 5 min on ice before fixation with ice-cold methanol for 12 min also on ice. Methanol was discarded and cells were washed 3 × 5 min with PBS at RT. All the following steps were performed using the Duolink PLA Reagents (DUO92001, DUO92005, DUO92008, DUO82049; Sigma-Aldrich) and according to the manufacturer's protocol. The blocking solution was applied for 1 h in a humidified incubator at 37°C, followed by antibody incubation overnight at 4°C under humid conditions. The following antibodies were used at the indicated dilutions: anti-MDC1 (Mouse monoclonal, 1:300, ab50003, RRID: AB_881103; Abcam), anti-HA (Rabbit polyclonal, 1:250, ab9110, RRID: AB_307019; Abcam). Next, coverslips were washed 2 × 5 min with wash buffer A at RT and the PLUS and MINUS PLA probes were applied for 1 h at 37°C in a humidified incubator. Cells were washed again 2 × 5 min with wash buffer A at RT and incubated with the ligation solution for 30 min in a humidified incubator at 37°C. After washing the coverslips 2 × 5 min with wash buffer A at RT, the amplification solution was applied for 100 min at 37°C under humid conditions. Final washes were carried out at RT, 2 × 10 min with wash buffer B and for 1 min with 0.01X wash buffer B. Coverslips were then mounted on glass microscopy slides using VECTASHIELD PLUS Antifade Mounting Medium with DAPI.

## Widefield and confocal microscopy

Widefield images of the PLA experiment were acquired with a Leica DMI6000B inverted fluorescence microscope, equipped with Leica K5 sCMOS fluorescence camera (16-bit, 2,048 × 2,048 pixel, 4.2 MP) and Las X software version 3.7.2.22383. An HC Plan Apochromat 40X/0.95 phase contrast air objective and Plan Apochromat 100X/1.40 phase contrast oil-immersion objective were used for image acquisition. For triple-wavelength emission detection, we combined DAPI with EGFP and Texas Red.

Confocal images of the microlaser and IRIF experiments were acquired with a Leica SP8 confocal laser scanning microscope

coupled to a Leica DMI6000B inverted stand, with a 63X, 1.4-NA Plan Apochromat oil-immersion objective. For quadruple-wavelength emission detection, we combined DAPI with EGFP, Alexa Fluor 568 or TurboRFP and Alexa Fluor 700. The sequential scanning mode was applied, and the number of overexposed pixels was kept at a minimum. 5 z-sections were recorded with optimal distances based on Nyquist criterion, a resolution of 512 × 512 pixels and 8-bit depth.

For optimal representation in figures, maximum intensity projections were calculated using Fiji (Schindelin et al, 2012). Unprocessed grayscale tagged image files (TIFs) and maximum intensity projections of confocal z-stacks were exported from Fiji, followed by pseudo-colouring and adjustment of exposure or brightness/contrast in Affinity Photo V2 (Affinity Package V2, Serif Europe Ltd, affinity.serif.com/en-us/photo/). For maximum data transparency and preservation, unprocessed grayscale images or maximum intensity projections were imported as smart objects and adjustments were performed using adjustment layers. Processed images were saved as multilayer PSD files.

### Image quantification

Adjusted volume (intensity) of Western blot bands was calculated with Image Lab (Bio-Rad, Image Lab Software Version 6.1, Bio-Rad Laboratories, Inc.) for every biotinylated p-peptide from the MDC1-[p]TQXF peptide pull-down from HeLa nuclear extracts, as a way of analyzing the binding affinity of either RNF8 or PELI1/2 for each pTQXF cluster. For quantitative assessment of protein colocalization in the IRIF experiment, maximum intensity projections of two channel confocal micrographs of single cells (EGFP and TurboRFP) were used for the SQUASSH analysis with the SQUASSH plugin (part of the MosaicSuite) for ImageJ and Fiji (Rizk et al, 2014; Leimbacher et al, 2019), imagej.net/Squassh. Widefield images of the PLA experiment were analyzed for quantification of the PLA signal using CellProfiler 4.0 (McQuin et al, 2018). Nuclei and GFP-positive regions were segmented with the intensity-based primary object detection module using the DAPI and GFP signals, respectively. GFP-positive nuclei were masked, and intensity of the PLA signal was measured on the respective channel and on the masked objects. The CellProfiler pipeline used is available upon request.

### Statistics and reproducibility

The appropriate statistical test was chosen as follows: to compare two populations, unpaired normal distributed data were tested with a two-tailed $t$ test (in case of similar variances) or with a two-tailed $t$ test with Welch's correction (in case of different variances). Unpaired non-continuous (count) data or non-normal distributed data were tested with two-tailed Mann-Whitney test (in case of similar variances) or with a two-tailed Kolmogorov–Smirnov test (in case of different variances). Three or more groups were analyzed by one-way ANOVA with Dunnett's correction for multiple comparison. In case of non-continuous (count) data or non-normal distribution, Kruskal-Wallis with Dunn's correction for multiple comparison was used.

All statistical analyses were performed with GraphPad Prism v9.00 (GraphPad Prism version 9.00 GraphPad Software, La Jolla California USA, graphpad.com). Sample sizes, $\alpha$ levels, and the statistical tests used are specified in the figure legends.

## Supplementary Information

## Acknowledgements

Imaging was performed with equipment maintained by the Center for Microscopy and Image Analysis, University of Zürich. Mass spectrometry was carried out at the Functional Genomics Centre of the University of Zürich. Cell sorting was carried out by the Flow Cytometry Core Facilities at the University of Zürich. X-ray data were collected at the Diamond Light Source, UK with the assistance of Simon Caulton. MJ Stewart is supported by the Biotechnology and Biological Sciences Research Council (BBSRC) and University of Birmingham funded Midlands Integrative Biosciences Training Partnership (MIBTP) BB/T00746X/1. SJ Smerdon is supported by a transitional award from the Francis Crick Institute which receives its core funding from Cancer Research UK (FC001003), the UK Medical Research Council (FC001003), and the Wellcome Trust (FC001003). The Stucki laboratory is supported by two project grants from the Swiss National Science Foundation (31003A_163141 and 310030_189141) and by the Kanton of Zürich.

### Author Contributions

M Torres Esteban: investigation, data curation, formal analysis, methodology, resources, visualization, and writing—review and editing.
MJ Stewart: investigation, data curation, formal analysis, methodology, resources, visualization, and writing—review and editing.
E Bragginton: investigation, methodology, and resources.
A Meroni: investigation, validation, and methodology.
A Pellizzari: investigation, validation, and methodology.
A Jeanrenaud: investigation, methodology, and data curation.
SJ Smerdon: conceptualization, formal analysis, funding acquisition, project administration, supervision, visualization, and writing—original draft.
M Stucki: conceptualization, formal analysis, funding acquisition, project administration, supervision, visualization, and writing—original draft.

### Conflict of Interest Statement

The authors declare that they have no conflict of interest.

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
