## [Reviewer comments · Life Science Alliance]

Life Science Alliance

MDC1 mediates Pellino recruitment to sites of DNA double-strand breaks

Monica Torres Esteban, Matthew Stewart, Eilis Bragginton, Alice Meroni, Annica Pellizzari, Alain Jeanrenaud, Stephen Smerdon, and Manuel Stucki

DOI: <https://doi.org/10.26508/lsa.202403074>

Corresponding author(s): Manuel Stucki, University of Zurich and Stephen Smerdon, University of Birmingham

Review Timeline:

Submission Date:	2024-10-07
Editorial Decision:	2024-10-07
Revision Received:	2025-01-29
Editorial Decision:	2025-02-13
Revision Received:	2025-02-18
Accepted:	2025-02-18

Transaction Report:

Please note that the manuscript was reviewed at *Review Commons* and these reports were taken into account in the decision-making process at *Life Science Alliance*.

Reviews

Review #1

****Summary:****

The manuscript by Esteban et al. provides compelling evidence that MDC1, in addition to its well-known interaction with the E3 ligase RNF8, also directly binds to the E3 ubiquitin ligases Pellino1 and Pellino2, both of which have recently emerged as important players in the DNA damage response. The authors convincingly show that MDC1 mediates the recruitment of Pellino1/2 to sites of DNA double-strand breaks (DSBs) via a phosphorylation-dependent interaction.

****Major Comments:****

1. **Competitive Binding Dynamics:** The manuscript should address whether RNF8 and Pellino1/2 compete for binding to phosphorylated MDC1. Understanding whether these interactions are competitive, cooperative, or mutually exclusive is crucial for deciphering the functional implications of MDC1's ability to interact with multiple E3 ligases.

2. **DNA Damage-Independent Interaction:** The observation that MDC1 and Pellino1/2 interact in the absence of DNA damage (as depicted in Figure 4A) is unexpected. While the authors note that MDC1 is phosphorylated to some degree in non-irradiated cells, this explanation does not fully clarify the mechanism or significance of this interaction in the absence of DNA damage. Since DNA damage typically enhances MDC1 phosphorylation, one would anticipate a corresponding increase in the interaction between MDC1 and Pellino1/2. Further investigation into this aspect is needed to better understand the context in which these interactions occur.

****Minor Comments:****

Figure 1E Labeling: There is a labeling error in Figure 1E; the labels should be corrected to "H2AX" and "pS139-H2AX."

2. Significance:

While the discovery of this interaction is intriguing, the manuscript would benefit from a more thorough exploration of its physiological significance.

Review #2

Evidence, reproducibility and clarity

The authors present a series of experiments demonstrating a physical interaction between the phosphorylated TQXF motifs of MDC1 and Pellino1 / Pellino 2 proteins (PELI1/PELI2). Both proteins are enriched in a peptide pull-down mass spectrometry experiment using HeLa nuclear extracts, and shown to bind preferentially to MDC1-pThr699 and -pThr765. These data are nicely supported by iso-thermal titration calorimetry experiments with a dissociation constant of 230nM determined for interaction with a synthetic pT765 phospho-peptide. The authors go on to determine an X-ray crystal structure of the FHA domain from PELI2 in complex with the MDC1-pThr765, to reveal the molecular determinants underpinning the interaction. Additional experiments, including immunoprecipitation, proximity ligation assay, and laser micro-irradiation support a phospho-specific interaction, driven by DNA damage, between MDC1 and PELI1/2 - and discount a direct interaction of PELI1/2 with the phosphorylated tail of histone H2AX as proposed previously in Ha et al (2019).

****Major comments:****

a) In all cases, RNF8 binds more tightly to the MDC1-pTQXF motifs, with the notable exception of pThr765; however, the affinity of RNF8 for the MDC1-pThr765 peptide has not been determined by ITC, which would be a useful comparator. Esp., given the question of how the authors envisage the interchange between binding to RNF8 vs PELI1/2?

b) Does Wing-II of PELI1/2 provide a secondary interaction pocket, capable of interacting with an extended bis-phosphorylated peptide? Is it highly conserved in terms of amino acid sequence?

c) Do the authors findings support the idea of PELI1 acting as part of a positive feedback mechanism that functions via PELI1-dependent ubiquitylation of NBS1?

d) The manuscript appears a little truncated, and does not explore the downstream (cellular) effects of disrupting the PELI1/2 - MDC1 interaction.

2. Significance:

Overall, the manuscript contains a robust and well-controlled set of experiments that confirm a direct interaction between the PELI1/2 protein and MDC1, dependent on prior phosphorylation of MDC1 (at its TQXF) sites in response to DNA damage. With that said, it is somewhat limited in scope, and does not explore the cellular consequences of breaking/interfering with the ability of PELI1/2 to interact with MDC1 and how this might be integrated into our understanding of the mammalian DNA damage response.

Review #3

1. Evidence, reproducibility and clarity:

The authors in this work provide compelling evidence to support a critical role of MDC1 as a molecular scaffold that dock Pellino1/2 at DSBs. Using an unbiased approach Pellino1/2, alongside RNF8, were identified as FHA-containing DNA damage responsive factors that bear affinity for an MDC1 TQXF phospho-peptide. Results from a series of interaction studies suggest that Pellino1/2 highly likely interact with MDC1 in a phosphorylation-dependent manner. Consistently, Pellino1/2 assembly at DSBs required MDC1 and its TQXF phosphorylation. Overall the work is of high quality and observations that highlight the MDC1-Pellino1/2 interaction are solid.

2. Significance:

The work offers an alternative perspective that details MDC1 as an important molecular scaffold that allows DSB targeting of Pellino1/2. I have only two minor comments -

1) As opposed to examining interaction between Pellino1/2 and the MDC1 fragment (Figure 4A), to unequivocally demonstrate that Pellino1/2 interacts with MDC1 in response to genotoxic stress the authors should show that Pellino1/2 interacts with full-length MDC1 but not its AQXF mutant. In this setting IR should further enhance the Pellino1/2-MDC1 interaction.

2) For visualisation purpose the authors should provide density map that shows the MDC1-pTQXF and the key interacting residues on the PELI2 FHA.

OPTIONAL:

Given the plausibility that Pellino1/2 may co-occupy MDC1 with NBS1 and RNF8 it would be of interest to examine if that is indeed the case. & Does Pellino1/2 expression affect RNF8-dependent ubiquitylation?

Review #4

1. Evidence, reproducibility and clarity:

In this manuscript by Torres Esteban et. al., the authors investigate how Pellino1/2 are recruited to chromatin surrounding DSBs. These two proteins contain the well-established FHA phospho-peptide binding domain that are present in other DDR chromatin binding such as RNF8. These two proteins were recently implicated as being involved in the DNA damage response, however, the exact phosphorylation-dependent recruitment method Pellino1/2 is not well defined.

The authors first attempt to identify proteins that bind to the multiple MDC1 TQXF motif, specifically when phosphorylated on the Threonine residue. In two independent experiments both Pellino1 and Pellino2 were pulled down with a phosphorylated MDC1 pTQXF peptide along with RNF8, as expected. They further show that this interaction between Pellino1/2 and the MDC1 is not specific to just a single MDC1 TQXF motif and that it requires a phosphorylated Threonine residue both from nuclear extracts and an in-vitro binding assay. The authors also show that Pellino1/2 cannot be pulled down from nuclear extracts using a pS139 H2AX peptide, as had been suggested by immunoprecipitation methods previously. Next, in Figure 2 the authors investigate if any of the 4 MDC1 TQXF motifs are preferentially bound by Pellino1/2 in comparison to RNF8. The authors further provide evidence for this phosphor-

dependent interaction with a crystal structure of Pellino2-FHA along with a pTQXF MDC1 peptide. They then show that while this Pel1/2-MDC1 interaction can be observed using immunoprecipitation of HA/Flag-MDC1, it appears to be IR-induced DNA damage independent with this method. Using PLA as a second approach they validate that Pellino1 and MDC1 can be found in close proximity within the nucleus in cells and that this interaction is enhanced upon IR-induced DNA damage. As a control they see no PLA foci in MDC1 KO U2OS cells. The authors further this notion that MDC1 is important for Pellino1/2 recruitment to DSBs with laser micro-irradiation and see that GFP-Pellino1 is recruited to DNA damage in an MDC1-dependent manner. They further show that this MDC1-dependent recruitment of Pellino1 requires the TQXF motif of MDC1 as mutation of all 4 TQXF residues to AQXF abolished Pellino1 recruitment to laser induced DNA damage. The authors then show that this same AQXF mutant is also devoid of Pellino1 recruitment ability using IR.

Overall, the manuscript is very clear and concise and presents a reasonable model that MDC1 pTQXF motifs are required for recruitment of Pellino1/2 to DSBs. The explanations of data and figures, including the rationale behind experiments are easily understood and well presented. There are a few points that should be addressed.

****Major Comments:****

1. What were the other proteins that came down with the phospho-MDC1 pTQXF peptide pulldown, and what were the statistics of the pulldowns from the mass spectrometry experiments (i.e. number of distinct peptides, coverage, etc, for all of the interacting proteins that were identified. I was not able find a table showing all of the other interactors (Table S1? Not included in the files). These data are critical and need to be presented in the manuscript.

2. Given that the FHA domain structure primarily recognizes the three amino acids C-terminal to the pThr residue, why do the authors think that the measured KD for the pT765 site (pTQPF) is 5-fold lower than the measured KD for the pT752 site (pTQPF)? Please provide some rationale for this in the text. (See point 4 below as well)

3. In Figure 2, the pT699 peptide binds strongest to RNF8 and PELI1/2. Is this a consequence of other factors (conformational or allosteric changes, or other modifications on these proteins from the nuclear extracts) in the full length proteins compared to the isolated FHA domains? If the authors repeat this pulldown experiment using only the isolated FLAG- or GST-tagged FHA domains, do they also get the same result? The data in Table T1 suggests this will not be the case. Please confirm this, and if the data and discordant, please comment on the difference in full-length RNF8 and PELI1/2 binding from the HeLa nuclear extracts versus the isolated FHA domains in the text.

4. In Figure 3, I am confused by the reference to Asp+5. The sequence around pT765 is pTQPFDT, so isn't the critical Asp in the +4 position? Is this interaction sufficient to account for the 5-fold tighter binding of this site than to the pT752 site? Or is this higher affinity a consequence of the pT-1 Glu residue, or both? Perhaps showing a full side chain interaction map would help here.

5. It would be helpful if the authors looked at RNF8 levels in the FLAG Ips shown in Figure 4A. It looks like DNA damage slightly reduced PELI1/2 binding which might be accounted for by RNF8 competition.

6. Do the authors have any data, or can they speculate, on whether the entire MDC1 699-770 region can simultaneously bind to both RNF8 and PELI1/2 though different sites at the same time on a single MDC1 molecule or is the RNF8-PELI1/2 co-localization from adjacent MDC1 molecules? While clearly not required, it would be interesting to ask whether transfecting MDC1-knockout cells with variations of MDC1 containing 1, 2, or 3 Thr-Ala mutations at these sites compromises co-localization and DNA damage responses? Can co-transfection of two constructs containing distinct single TQXF sites - one optimized for RNF8 binding and one optimized for PELI1/2 binding, can rescue the co-localization phenotype.

7. What effect does the combined knock-down of PELI1/2 have on some aspect of DNA damage signaling or cell cycle progression? Is the repair/loss of gH2AX or 53BP1 foci delayed if PELI1/2 are absent?

8. Figure 4A shows that the interaction between MDC1 and Pellino1/2 is independent of IR-induced DNA damage as observed using immunoprecipitation. 4B shows that PLA foci between MDC1 and Strep-HA-GFP-Pellino1 can be observed in undamaged cells and PLA foci are increased when observed 3h post 3Gy IR damage. Is this discrepancy explainable by differences between PELI1/2 recruitment given that the antibody in 4A used recognizes both isoforms but the PLA in 4B is specific to Pellino1? It would be nice to see if there are any differences between Pellino1/2 by

performing IP +/- DNA damage using exogenously expressed Pellino1 or 2 variants. Additionally, repeating PLA in 4B with a Pellino2 construct would help address this question.

9. Similar to the comment above, a further investigation into if there are any differences in recruitment between Pellino1/2 using the experimental systems used in Figure 5 would greatly support the model presented, particularly since the crystal structure presented in Figure 3 is with Pellino2. Is there any rationale for this based upon amino acid sequence differences between Pellino1/2?

10. Using the experimental approaches in Figures 4 & 5 and the insight from the crystal structure in Figure 3, making point mutations in Pellino1/2 that would be predicted to disrupt the phospho-specific interaction would provide further confidence in the model proposed.

****Minor Comments:****

1. Could the authors please comment on if Pellino3a/b came down in the initial IP experiment and if not, is there potentially an explanation based upon sequence similarities/differences as inferred by the crystal structure of Pellino2?

2. Is there a rationale for why only single TQXF motifs were tested in isolation for binding in 2A? Given the proximity of the TQXF sites one could imagine that 2 or 3 TQXF sites being simultaneously phosphorylated could provide an even better peptide for Pellino1/2 FHA binding.

3. In Figure 2B, how is the data "Adjusted"?

4. Table T2 is provided twice, but the Table of co-precipitating proteins from the mass spectrometry experiments is not shown.

****REFEREE CROSS-COMMENTS****

It seems that most of us hit on the same set of core issues - whether the MDC stretch of 4 pTQxF motifs simultaneously recruits RNF8 and Pellino1/2 to the same MDC1 molecules or not, a few more mutational studies, and whether the Pellino recruitment is important in events downstream of MDC1 binding. This latter issue could entail significantly more work, though its inclusion would likely influence which of the journals that use Review Commons was interested in pursuing the paper. A purely biochemical study is fine, at least in my opinion, depending on the type of journal, but adding some functional DNA damage context and relevant cell biology would bring the work to a deeper level. Just my thoughts, of course....

2. Significance:

The general significance of the data presented by Torres Esteban et. al., provide a mechanistic understanding of how Pellino1/2 are recruited to DNA damage, a role that has previously not been characterized. While this manuscript does not delve into the role of Pellino1/2 downstream of its recruitment, much of this has been investigated by a paper by Ha et. al., 2019 as cited in this manuscript. In fact, the Ha et. al., propose that Pellino1/2 directly interacts with pS139 H2AX. Torres Esteban et. al., convincingly show that this is likely not the case and provide robust biochemical and cellular evidence to support their mechanistic model, very nicely contributing to the body of knowledge regarding Pellino1/2 in the DDR. The audience that this paper would be important for would be those in the DNA damage field or structural biologists interested in phospho-peptide binding domains.

Our expertise is in DNA damage signaling, FHA and other protein modular domains, and determination of protein structure by X-ray crystallography.

October 7, 2024

Re: Life Science Alliance manuscript #LSA-2024-03074-T

Dr. Manuel Stucki
University of Zurich
Department of Gynecology USZ Wagistrasse 14
Schlieren 8952

Dear Dr. Stucki,

Thank you for submitting your manuscript entitled "MDC1 mediates Pellino recruitment to sites of DNA double-strand breaks" to Life Science Alliance. We invite you to re-submit the manuscript, revised according to your Revision Plan.

Thank you for this interesting contribution to Life Science Alliance. We are looking forward to receiving your revised manuscript.

Sincerely,

B. MANUSCRIPT ORGANIZATION AND FORMATTING:

Point-by-point Response to Reviewer's Comments

We would like to thank the reviewers for their thorough and careful evaluation of our manuscript and their suggestions for improving the manuscript by incorporating textual and experimental revisions. We have experimentally addressed some of the issues raised (see below), but we were unfortunately not able to address all of them, mainly because of technical limitations described in the specific responses below, as well as resource limitations, especially with regards to staff resources. In the following sections, we provide point-to-point responses to the reviewer's comments, which are in *italic*. For simplicity, we combined similar issues raised by different reviewers and responded to them together.

Reviewer #1: The manuscript should address whether RNF8 and Pellino1/2 compete for binding to phosphorylated MDC1. Understanding whether these interactions are competitive, cooperative, or mutually exclusive is crucial for deciphering the functional implications of MDC1's ability to interact with multiple E3 ligases.

Reviewer #2: [...] given the question of how the authors envisage the interchange between binding to RNF8 vs PELI1/2?

Reviewer #3: Given the plausibility that Pellino1/2 may co-occupy MDC1 with NBS1 and RNF8 it would be of interest to examine if that is indeed the case. & Does Pellino1/2 expression affect RNF8-dependent ubiquitylation?

Reviewer #4: Do the authors have any data, or can they speculate, on whether the entire MDC1 699-770 region can simultaneously bind to both RNF8 and PELI1/2 though different sites at the same time on a Ber1076!89AMGs-! single MDC1 molecule or is the RNF8-PELI1/2 co-localization from adjacent MDC1 molecules?

It is not trivial to address this question directly. RNF8 and PELI1/2 bind to the pTQXF sites in MDC1 in vitro and the binding affinities are overall very similar (Huen, et al., (2007). RNF8 Transduces the DNA-Damage Signal via Histone Ubiquitylation and Checkpoint Protein Assembly. *Cell* 131, 901-914; and this study). Therefore, they will inevitably compete for the sites but there is no reason to expect overall exclusivity of one over the other unless one is substantially overexpressed.

To address this issue experimentally in the revised version of this manuscript we tested if PELI1 and RNF8 can simultaneously be recruited to sites of DSBs. For this, we overexpressed GFP-tagged RNF8 and RFP-tagged PELI1 and assessed their co-localization with MDC1 in foci at sites of DSBs. The results showed that both overexpressed GFP-RNF8 and RFP-PELI1 did co-localize with each other and with MDC1 in foci at sites of DNA damage, thus indicating that RNF8 and PELI1/2 can simultaneously bind to MDC1 pTQXF motifs in cells, even under overexpression conditions (new Fig S3).

Reviewer #1: DNA Damage-Independent Interaction: The observation that MDC1 and Pellino1/2 interact in the absence of DNA damage (as depicted in Figure 4A) is unexpected. While the

authors note that MDC1 is phosphorylated to some degree in non-irradiated cells, this explanation does not fully clarify the mechanism or significance of this interaction in the absence of DNA damage. Since DNA damage typically enhances MDC1 phosphorylation, one would anticipate a corresponding increase in the interaction between MDC1 and Pellino1/2. Further investigation into this aspect is needed to better understand the context in which these interactions occur.

Reviewer #3: As opposed to examining interaction between Pellino1/2 and the MDC1 fragment (Figure 4A), to unequivocally demonstrate that Pellino1/2 interacts with MDC1 in response to genotoxic stress the authors should show that Pellino1/2 interacts with full-length MDC1 but not its AQXF mutant. In this setting IR should further enhance the Pellino1/2-MDC1 interaction.

Reviewer #4: It would be helpful if the authors looked at RNF8 levels in the FLAG Ips shown in Figure 4A. It looks like DNA damage slightly reduced PELI1/2 binding which might be accounted for by RNF8 competition.

Reviewer #4:

5. It would be helpful if the authors looked at RNF8 levels in the FLAG Ips shown in Figure 4A. It looks like DNA damage slightly reduced PELI1/2 binding which might be accounted for by RNF8 competition.

8. Figure 4A shows that the interaction between MDC1 and Pellino1/2 is independent of IR-induced DNA damage as observed using immunoprecipitation. 4B shows that PLA foci between MDC1 and Strep-HA-GFP-Pellino1 can be observed in undamaged cells and PLA foci are increased when observed 3h post 3Gy IR damage. Is this discrepancy explainable by differences between PELI1/2 recruitment given that the antibody in 4A used recognizes both isoforms but the PLA in 4B is specific to Pellino1? It would be nice to see if there are any differences between Pellino1/2 by performing IP +/- DNA damage using exogenously expressed Pellino1 or 2 variants.

We have performed a few additional co-IP experiments, using MDC1 wild type and AQXF mutants, both in the presence and absence of DNA damage. These new experiments showed a partial dependency of the MDC1-PELI interaction on exogenous DNA damage, because the PELI1/2 signal is stronger in co-IP reactions performed in extracts of irradiated cells (Fig 4A). We also performed co-IP experiments in cells that were transfected with a mutant MDC1 that had the Thr residue in all four TQXF motifs replaced by Ala (AQXF). This mutant did not significantly interact with PELI1/2, irrespective of DNA damage. Reviewer #3 suggested using full-length MDC1 but this has historically been an unsuccessful approach in other studies since full-length protein is tightly bound to damaged chromatin and requires high-salt extraction which inevitably disrupts phospho-dependent binding. However, using a truncated fragment (MDC1 800) lacking the C-terminal BRCT domain at least partly solves this problem and has been previously used by ourselves (Jungmichel et al., (2012). The molecular basis of ATM-dependent dimerization of the Mdc1 DNA damage checkpoint mediator. *Nucleic acids research* 40, 3913-3928) and by other laboratories (see e.g. Mailand, et al., (2007). RNF8 ubiquitylates histones at DNA double-strand breaks and promotes assembly of repair proteins. *Cell* 131, 887-900.) to address similar questions in other contexts.

Reviewer #3: For visualisation purpose the authors should provide density map that shows the MDC1-pTQXF and the key interacting residues on the PELI2 FHA.

A density map is already shown in Fig 3A. We have added a cartoon of the major protein-peptide contacts to this figure to increase the clarity (new Fig S2).

Reviewer #1:

Figure 1E Labeling: There is a labeling error in Figure 1E; the labels should be corrected to "H2AX" and "pS139-H2AX."

This mistake has been corrected.

Reviewer #2:

Does Wing-II of PELI1/2 provide a secondary interaction pocket, capable of interacting with an extended bis-phosphorylated peptide? Is it highly conserved in terms of amino acid sequence?

Wing II is highly conserved in PELI1/2 but, as we say, doesn't make direct contacts with the MDC1 pTQXF motif. The area of positive potential could, in principle bind a second phosphosite but, as shown by the Ferguson lab (Huoh and Ferguson (2014). The Pellino E3 Ubiquitin ligases Recognize Specific Phosphothreonine Motifs and Have Distinct Substrate Specificities. *Biochemistry* 53, 4946-4955), Asp seen in the +3 position of some previous FHA targets (eg Rad53 FHA1) has the weakest affinity for PELI of all the peptide variants used in that study.

Reviewer #2:

Do the authors findings support the idea of PELI1 acting as part of a positive feedback mechanism that functions via PELI1-dependent ubiquitylation of NBS1?

We did not address this question experimentally. Therefore, we are not able to either confirm or dismiss the model proposed by Ha et al.

Reviewer #4:

What were the other proteins that came down with the phospho-MDC1 pTQXF peptide pulldown, and what were the statistics of the pulldowns from the mass spectrometry experiments (i.e. number of distinct peptides, coverage, etc, for all of the interacting proteins that were identified. I was not able find a table showing all of the other interactors (Table S1? Not included in the files). These data are critical and need to be presented in the manuscript.

We apologize for this mistake. The full list of proteins that came down with the phospho-MDC1 pTQXF peptide is now provided in Table 1.

Reviewer #4:

In Figure 3, I am confused by the reference to Asp+5. The sequence around pT765 is pTQPFDT, so isn't the critical Asp in the +4 position? Is this interaction sufficient to account for the 5-fold tighter binding of this site than to the pT752 site? Or is this higher affinity a

consequence of the pT-1 Glu residue, or both? Perhaps showing a full side chain interaction map would help here.

This is a mistake in the labeling of the Asp residue, which has been corrected. The pT +4 Asp likely contributes to the difference in affinities as it is juxtaposed with an area of positive electrostatic potential. This feature has now been added to the description of the phosphopeptide complex. A full interaction map is now provided (new Fig S2).

Reviewer #4:

Given that the FHA domain structure primarily recognizes the three amino acids C-terminal to the pThr residue, why do the authors think that the measured KD for the pT765 site (pTQPF) is 5-fold lower than the measured KD for the pT752 site (pTQPF)? Please provide some rationale for this in the text. (See point 4 below as well)

Previous studies on canonical FHA domains have highlighted a preference for residues in the pT+3 with less significant contributions from the +1 and +2 positions. In the case of the variant FHA domain in PELI2, our structure suggests that the difference in affinity alluded to by the referee is likely attributable to an electrostatic interaction between the Asp+4 in the pT765 site that is absent in the pT752 motif. This discussion has been added.

Reviewer #4:

7. What effect does the combined knock-down of PELI1/2 have on some aspect of DNA damage signaling or cell cycle progression? Is the repair/loss of gH2AX or 53BP1 foci delayed if PELI1/2 are absent?

Unfortunately, we are not able to address this point, which was also raised by the other reviewers. The reason is that we haven't been able to efficiently deplete PELI1 and 2 from cells, which would be essential to perform the proposed loss-of-function analysis. We attempted both siRNA (Dharmacon pool) mediated knock-down and CRISPR/Cas9 mediated knock-out of PELI1 and 2 but neither approach was successful. The siRNA pool was not efficiently depleting PELI1 and 2. We were also unable to derive single PELI1 knock-out clones or PELI1 and 2 double knock-out clones from the CRISPR/Cas9 experiments. Monica Torres Esteban (co-first author on this study) spent close to a year trying to establish a cellular system for loss-of-function analysis, without success. Therefore, we had to abandon these plans. Since Monica has finished her PhD and left the Stucki lab in April 2024, we also currently lack the staff resources to revisit this aspect.

Reviewer #4:

9. Similar to the comment above, a further investigation into if there are any differences in recruitment between Pellino1/2 using the experimental systems used in Figure 5 would greatly

support the model presented, particularly since the crystal structure presented in Figure 3 is with Pellino2. Is there any rationale for this based upon amino acid sequence differences between Pellino1/2?

In response to this suggestion, we reiterate that we see no significant difference in binding of PELI1 and 2 either to MDC1 sites or even to a high affinity IRAK1 site by ITC titration (differences within the ranges reported by Huoh and Ferguson (2014). The Pellino E3 Ubiquitin Ligases Recognize Specific Phosphothreonine Motifs and Have Distinct Substrate Specificities. *Biochemistry* 53, 4946-4955). The two proteins are highly conserved (82% ID over the entire protein) and, importantly, all residues in contact with the peptide motif in our crystal structure are identical. In the light of these observations, we feel that the substantial additional work required to produce mammalian cell expression constructs, derive stable cell lines and carry out the same set of fluorescence measurements for PELI2 is not justified and would add little to the narrative since the relative importance/significance of the two paralogues in the context of MDC1 binding is not under consideration in the current paper. The high degree of similarity at the sequence, structural and in vitro biochemical level are perhaps not sufficiently emphasized and will provide the appended structure-based sequence alignment analysis to illustrate this better (Figure 1).

Reviewer #4:

10. Using the experimental approaches in Figures 4 & 5 and the insight from the crystal structure in Figure 3, making point mutations in Pellino1/2 that would be predicted to disrupt the phospho-specific interaction would provide further confidence in the model proposed.

Again, we understand the basis of the comment but rather feel that these experiments would add little. We have shown that TQXF to AQXF mutations in MDC1 lead to defective PELI1 interaction and we are now showing by additional co-IP experiments (see above) that the AQXF mutant also doesn't interact with PELI1/2 anymore. Ha et al. have previously shown that disruption of the PELI FHA domain abrogates recruitment to damage sites (Ha et al., (2019). Pellino1 regulates reversible ATM activation via NBS1 ubiquitination at DNA double-strand breaks. *Nature Communications* 1-18) and we have now shown biochemically and structurally a direct interaction with the MDC1 phospho-motifs. Mutating key amino acids in PELI FHA domains such as Ser135 and Arg104 will thus not provide further essential confidence because these amino acids are conserved in all known FHA domains and are always essential for the interaction with their corresponding phosphorylated binding partners. Indeed, the Ferguson lab has already shown that mutating R104, R106 and R131 in PELI1, 2 and 3a respectively is sufficient to abolish interactions with phosphopeptides derived from a canonical target IRAK1 (Huoh and Ferguson (2014). The Pellino E3 Ubiquitin Ligases Recognize Specific Phosphothreonine Motifs and Have Distinct Substrate Specificities. *Biochemistry* 53, 4946-4955): Fig1) and that this effect is recapitulated with IRAK1 in cells.

FHA domain

RING-like domain

Figure 1: Sequence alignment of human PELI1 and PELI2. Identical amino acids are marked red, similar ones are colored red on white background. Non-conserved amino acids are colored black on white background. FHA domain and RING-like domain are indicated. Blue asterisks indicate amino acid residues that directly contact the phosphopeptide in the X-ray structure. Green asterisks indicate amino acid residues that are interacting with the +3F.

February 13, 2025

RE: Life Science Alliance Manuscript #LSA-2024-03074-TR

Dr. Manuel Stucki
University of Zurich
Department of Gynecology USZ Wagistrasse 14
Schlieren 8952

Dear Dr. Stucki,

Thank you for submitting your revised manuscript entitled "MDC1 mediates Pellino recruitment to sites of DNA double-strand breaks". We would be happy to publish your paper in Life Science Alliance pending final revisions necessary to meet our formatting guidelines.

- please be sure that the authorship listing and order is correct
- please add ORCID ID for the secondary corresponding author -- they should have received instructions on how to do so
- please add the Twitter/X and Bluesky handles of your host institute/organization as well as your own or/and one of the authors in our system
- please be sure that the authorship listing and order are correct and match the system and manuscript file
- please consult our manuscript preparation guidelines <https://www.life-science-alliance.org/manuscript-prep> and make sure your manuscript sections are in the correct order -- Title page, Summary blurb, Abstract, Introduction, Results & Discussion, Materials and Methods, Data Availability, Acknowledgements, Author contributions, Conflict of interest, References, Figure legends, Tables, and their legends
- please use the [10 author names, et al.] format in your references (i.e. limit the author names to the first 10)
- there are call-outs for non-existing panels and tables, please correct (Figure 2C, Table 3, Table S3)
- please add callouts for Figures S1A-B and S3 to your main manuscript text

LSA now encourages authors to provide a 30-60 second video where the study is briefly explained. We will use these videos on social media to promote the published paper and the presenting author (for examples, see <https://docs.google.com/document/d/1-UWCfbE4pGcDdcgzcmiuJl2XMBJnxKYeqRvLLrLSo8s/edit?usp=sharing>). Corresponding or first-authors are welcome to submit the video. Please submit only one video per manuscript. The video can be emailed to contact@life-science-alliance.org

A. FINAL FILES:

B. MANUSCRIPT ORGANIZATION AND FORMATTING:

Sincerely,

Reviewer #1 (Comments to the Authors (Required)):

The authors have adequately addressed my concerns.

Reviewer #2 (Comments to the Authors (Required)):

The authors answered all my questions and clarified the manuscript. I have no further concerns, and I will look forward to seeing the paper published.

Reviewer #3 (Comments to the Authors (Required)):

I believe the authors have responded satisfactorily to all the comments, points and suggestions made in the previous review cycle. I therefore have no additional points to raise.

February 18, 2025

RE: Life Science Alliance Manuscript #LSA-2024-03074-TRR

Prof. Manuel Stucki
University of Zurich
Wagistrasse 14
Schlieren 8952
Switzerland

Dear Dr. Stucki,

Thank you for submitting your Research Article entitled "MDC1 mediates Pellino recruitment to sites of DNA double-strand breaks". It is a pleasure to let you know that your manuscript is now accepted for publication in Life Science Alliance. Congratulations on this interesting work.

DISTRIBUTION OF MATERIALS:

Again, congratulations on a very nice paper. I hope you found the review process to be constructive and are pleased with how the manuscript was handled editorially. We look forward to future exciting submissions from your lab.

Sincerely,
